# MYOD1 functions as a clock amplifier as well as a critical co-factor for downstream circadian gene expression in muscle

Brian A Hodge[1†‡], Xiping Zhang[1†], Miguel A Gutierrez-Monreal[1], Yi Cao[2], David W Hammers[3], Zizhen Yao[4], Christopher A Wolff[1], Ping Du[1], Denise Kemler[1], Andrew R Judge[5], Karyn A Esser[1]*

[1]Department of Physiology and Functional Genomics, University of Florida, Gainesville, United States; [2]Department of Bioinformatics and Computational Biology, Genentech Inc, South San Francisco, United States; [3]Department of Pharmacology and Therapeutics, University of Florida Health Science Center, Gainesville, United States; [4]Allen Institute for Brain Science, Seattle, United States; [5]Department of Physical Therapy, University of Florida Health Science Center, Gainesville, United States

*For correspondence: kaesser@ufl.edu

†These authors contributed equally to this work

Present address: ‡Buck Institute for Research on Aging, Novato, United States

**Abstract** In the present study we show that the master myogenic regulatory factor, MYOD1, is a positive modulator of molecular clock amplitude and functions with the core clock factors for expression of clock-controlled genes in skeletal muscle. We demonstrate that MYOD1 directly regulates the expression and circadian amplitude of the positive core clock factor *Bmal1*. We identify a non-canonical E-box element in *Bmal1* and demonstrate that is required for full MYOD1-responsiveness. Bimolecular fluorescence complementation assays demonstrate that MYOD1 colocalizes with both BMAL1 and CLOCK throughout myonuclei. We demonstrate that MYOD1 and BMAL1:CLOCK work in a synergistic fashion through a tandem E-box to regulate the expression and amplitude of the muscle specific clock-controlled gene, Titin-cap (*Tcap*). In conclusion, these findings reveal mechanistic roles for the muscle specific transcription factor MYOD1 in the regulation of molecular clock amplitude as well as synergistic regulation of clock-controlled genes in skeletal muscle.

DOI: https://doi.org/10.7554/eLife.43017.001

## Introduction

Circadian rhythms are repetitive ~24 hr cycles that allow organisms to temporally align behavioral, biochemical and physiological processes with daily environmental changes (*Vitaterna et al., 2001*; *Panda et al., 2002*; *Bhadra et al., 2017*). These rhythms exist in virtually all mammalian cells and are generated by transcriptional/translational feedback loops referred to as the molecular-clock (*Partch et al., 2014*; *Tataroglu and Emery, 2015*; *Takahashi, 2016*). The positive limb of the core clock is comprised of the PAS domain containing basic-Helix-Loop-Helix factors (PAS-bHLH) core clock factors Brain and Muscle Arnt-Like 1 (*Bmal1*) and Circadian Locomotor Output Clocks Kaput (*CLOCK*). These factors heterodimerize and bind to the DNA at E-box elements where they generate circadian transcription oscillations through rhythmic recruitment of histone acetylases, co-factors, and components of the transcriptional complex (*King et al., 1997*; *Bunger et al., 2000*; *Partch et al., 2014*). In addition to keeping time, the core molecular clock factors regulate the expression of downstream clock-controlled genes (CCGs), many of which encode master transcriptional regulators and rate-limiting enzymes in key biochemical pathways (*Bozek et al., 2007*; *Bozek et al., 2009*).

Although the core molecular clock components are expressed in the majority of cell-types throughout the body, CCGs are expressed in a very tissue-specific fashion (*Storch et al., 2002*; *Zhang et al., 2014*; *Mure et al., 2018*). This temporal regulation of tissue-specific gene programs allows for the timing of organ and cell-type specific processes that help maintain physiological homeostasis within each tissue and across multiple organ systems throughout the day (*Bozek et al., 2009*; *Korenčič et al., 2015*). The transcriptional mechanisms by which the core clock factors regulate tissue-specific genes are not fully understood. Recent studies have begun to identify lineage-specific transcriptional regulators that co-localize with molecular clock components at cis-regulatory elements located within tissue-specific promoter and enhancer regions (*Bozek et al., 2007*; *Dufour et al., 2011*; *Korenčič et al., 2012*; *Perelis et al., 2015*). To date, factors within the liver, hippocampus, pancreas have previously been reported, however a muscle-specific transcriptional regulator has yet to be defined.

In skeletal muscle the bHLH transcription factor MYOD1 drives myogenic gene expression by recruiting co-factors and the transcriptional machinery to muscle-specific gene promoters (*Rudnicki et al., 1993*; *Polesskaya et al., 2001*; *Fong and Tapscott, 2013*; *Buckingham and Rigby, 2014*). MYOD1 is often referred to as the 'master myogenic switch' as it is required for muscle cell differentiation and is capable of converting non-muscle cells into a muscle lineage (*Davis et al., 1987*; *Tapscott et al., 1988*). In adult skeletal muscle, BMAL1:CLOCK target the core-enhancer element (CE) located 20 kb upstream of the *Myod1* start site to promote the circadian expression of MYOD1 (*Andrews et al., 2010*; *Zhang et al., 2012*). We previously reported that MYOD1-CE mice, that only lack the upstream CE region, display significant declines in the circadian amplitude of the core clock genes *Bmal1* and *Per2* (*Zhang et al., 2012*), suggesting MYOD1 may modulate core clock gene expression in skeletal muscle.

Herein, we sought to address two questions: 1) Does MYOD1 transcriptionally regulate core molecular clock genes? And 2) Does MYOD1 work with the core clock components to regulate the circadian expression of muscle specific genes? We found that MYOD1 binds to an intronic enhancer within the *Bmal1* promoter and functions to transcriptionally regulate *Bmal1* expression. Using both In vivo and In vitro approaches we determined that MYOD1 serves to enhance the amplitude of *Bmal1* expression creating a feed-forward regulatory loop between *MyoD1* and the core clock gene, *Bmal1* in skeletal muscle. We also found that MYOD1 works in a synergistic fashion with BMAL1:CLOCK to amplify the circadian expression of a muscle-specific, clock-controlled gene, *Titin-cap* (*Tcap*). Co-localization studies demonstrated that MYOD1, BMAL1, and CLOCK are in close proximity within myonuclei. *Tcap* promoter analysis uncovered that MYOD1 and BMAL1 target a tandem E-box and that both Eboxes are required for the circadian regulation. These findings identify a novel role for MYOD1 as a clock amplifier and highlight synergistic interactions among core the clock factors, BMAL1:CLOCK and MYOD1 in regulating downstream clock-controlled gene expression in skeletal muscle.

## Results

### Characterization of MYOD1 binding sites in adult skeletal muscle

We first noted that expression of the core clock genes *Bmal1* and *Per2* were dampened in skeletal muscle of mice in which circadian expression of *MyoD1* was abolished (MYOD1-CE mice), which suggested that MYOD1 may function as an upstream transcriptional regulator of the molecular clock (*Zhang et al., 2012*). To address these findings we performed a MYOD1 ChIP-Seq experiment with adult skeletal muscle from male C57BL/6J mice. We identified 12,343 MYOD1 binding sites on 7751 genes using very stringent statistics for calling peaks to minimize false positives due to our lack of a preimmune serum control (*Supplementary file 1*). We compared the list of genes bound by MYOD1 to a list of circadian genes identified from a high resolution time-series collection in skeletal muscle (*Zhang et al., 2014*). Of the 1454 circadian mRNA transcripts in skeletal muscle (JTK_CYCLE p-value < 0.03: *Supplementary file 2*) we found that approximately 30% (536 genes, *Supplementary file 3*) are directly targeted by MYOD1 (*Figure 1A*) (*Zhang et al., 2014*). Gene ontology (GO) enrichment analysis of these 536 circadian MYOD1 target genes revealed a significant enrichment for genes involved in muscle structure and development consistent with MYOD1's known function as a myogenic transcription factor (*Figure 1B*, *Supplementary file 4*).

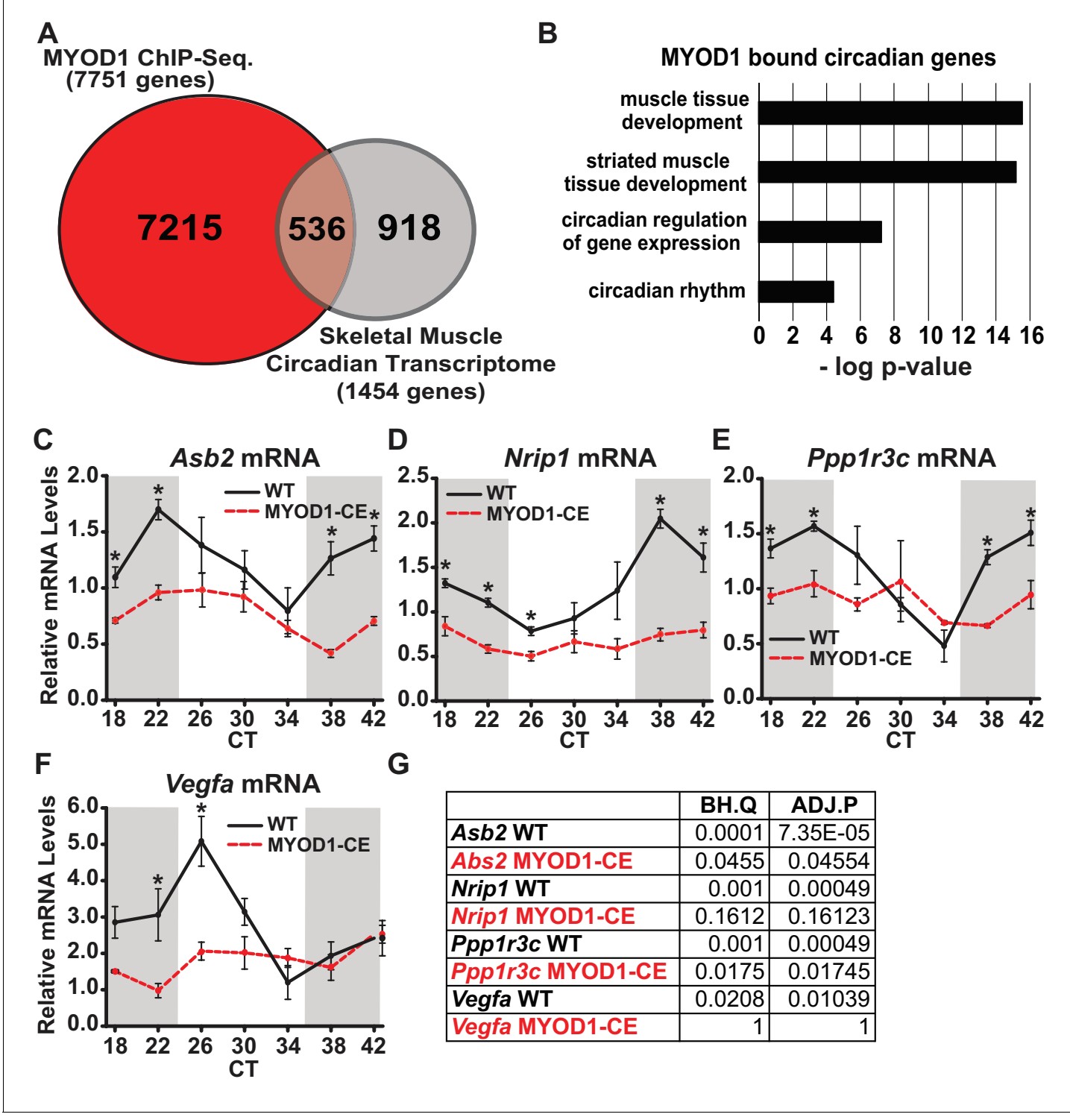

**Figure 1.** MYOD1 binding coverage on skeletal muscle circadian genes. (**A**) Overlap of genes bound by MYOD1 (red) and circadian genes (grey) in adult skeletal muscle (JTK_CYCLE p-value < 0.03). (**B**) Gene-ontology enrichment terms for MYOD1-bound, circadian genes in adult skeletal muscle. (**C-F**) Temporal mRNA expression profiles of MYOD1-bound, circadian genes in adult skeletal muscle from either MYOD1-CE (dotted red) or C57BL/6J (solid black, wildtype) mice. Dark shading indicates the relative dark/active phase as these mice were reared in DD at the time of collection. At each time-point RT-PCR expression values are displayed as average fold-change relative to the *Rpl26* house-keeping gene ± SEM (n = 3). Relative gene expression was calculated by the standard curve method. Results were analyzed with one-way ANOVA comparing WT vs. MYOD1-CE, * indicates a p-value less than 0.05. (**G**) JTK_CYCLE statistics for the RT-PCR results corresponding to the temporal expression values in C-F. 'BH.Q' column reports false discover rates and 'ADJ.P' reports adjusted p-values.

*Figure 1 continued on next page*

*Figure 1 continued*

DOI: https://doi.org/10.7554/eLife.43017.002

To futher investigate MYOD1 as a regulator of downstream circadian gene expression we selected a subset of the MYOD1-bound circadian target genes and evaluated their temporal expression profiles in skeletal muscle from the MYOD1-CE mice. We indentified target genes for our analysis by querying a publically available MYOD1-CE expression dataset for circadian genes that are also MYOD1 targets to test if they were down-regulated in MYOD1-CE muscle tissue (*Supplementary file 5*). The genes included in this analysis were the muscle-growth regulator *Asb2* (*Davey et al., 2016*), the clock-output gene *Nrip1* (*Poliandri et al., 2011*), a glycogen synthase regulator *Ppp1r3c* (*Montori-Grau et al., 2011*), and the angiogenesis factor *Vegfa* (*Arsic et al., 2004*). Interestingly, in muscle from MYOD1-CE mice these genes displayed altered temporal expression over time of day with a significant reduction in average expression (*Figure 1C–F*). When we analyzed the temporal gene expression profiles with the circadian statistical program, JTK_CYCLE, we found that *Vegfa* and *Nrip1* mRNAs were no longer circadian, and for *Ppp1r3c* and *Asb2* the p-values were notably higher. These findings are consistent with a model in which MYOD1 levels influence molecular clock transcriptional output in skeletal muscle.

## MYOD1 transcriptionally regulates the expression and circadian amplitude of *Bmal1*

Within the subset of MYOD1 bound circadian genes we found a significant enrichment for the biological process of 'circadian rhythm' (*Figure 1B*) and observed MYOD1 binding within gene-regulatory regions of multiple core clock genes (*Table 1*) including a large binding peak within the *Bmal1* locus (*Figure 2A*). MYOD1 over-expression significantly elevated endogenous BMAL1 protein levels by approximately 50% in C2C12 myotubes (*Figure 2B*), suggesting that *Bmal1* is under transcriptional regulation by MYOD1. We next tested the responsiveness of a *Bmal1* promoter luciferase reporter (*Bmal1P*-Luc) to over-expression of MYOD1 via Dual-Luciferase assays in C2C12 myotubes (*Figure 2C*) and isolated skeletal muscle primary myotubes (*Figure 2D*). For these experiments we co-transfected the *Bmal1P*-Luc reporter with an empty vector control (pGEM), a MYOD1 over-expression plasmid (MYOD1), or a MYOD1 mutant expression plasmid lacking a functional transactivation domain (MYOD1 3Δ56, MYOD1mut) (*Perry et al., 2001*). Over-expression of wildtype MYOD1 resulted in a significant enhancement in *Bmal1P*-Luc activity in comparison to the pGEM control or the MYOD1mut vector in both the C2C12 myotube or primary myotube cultures (*Figure 2C,D*). To directly test whether the *Bmal1* promoter can be transactivated by MYOD1 in adult skeletal muscle we electroporated *Bmal1P*-Luc into the tibialis anterior (TA) muscles with one leg co-transfected with pGEM control while the contralateral TA was co-transfected with a MYOD1 over-expression plasmid. We found that over-expression of MYOD1 was sufficient to significantly

**Table 1.** MYOD1 binding sites within core clock genes from skeletal muscle ChIP-Seq.

| Gene name | Peak score | Distance to TSS | Annotation | BH.Q | Adj.p | Phase | AMP |
|---|---|---|---|---|---|---|---|
| *Arntl* | 52.5 | −215 | promoter-TSS (NM_007489) | 6.4E-07 | 1.47E-10 | 23 | 0.8400075 |
| *Coq10b* | 17.5 | −52 | promoter-TSS (NM_001039710) | 1.04E-05 | 9.47E-09 | 9 | 0.5541172 |
| *Per3* | 15.3 | 84 | promoter-TSS (NM_001289878) | 1.46E-05 | 1.60E-08 | 10.5 | 0.4985951 |
| *Clock* | 17.5 | 206 | promoter-TSS (NM_001305222) | 2.96E-05 | 4.38E-08 | 1 | 0.2368383 |
| *Tef* | 23.3 | 397 | intron (NM_153484, intron 1 of 3) | 0.000268 | 9.96E-07 | 10.5 | 0.1759777 |
| *Nr1d2* | 27 | 276 | 5' UTR (NM_011584, exon 1 of 8) | 0.000359 | 1.49E-06 | 9.5 | 0.2442347 |
| *Ciart* | 19 | −384 | promoter-TSS (NM_001033302) | 0.000626 | 3.24E-06 | 9 | 1.0158296 |
| *Hlf* | 10.9 | 39 | promoter-TSS (NM_172563) | 0.001343 | 9.73E-06 | 11 | 0.3266197 |
| *Cry1* | 16.8 | −40 | promoter-TSS (NM_007771) | 0.026447 | 0.000699 | 19 | 0.0611011 |

DOI: https://doi.org/10.7554/eLife.43017.004

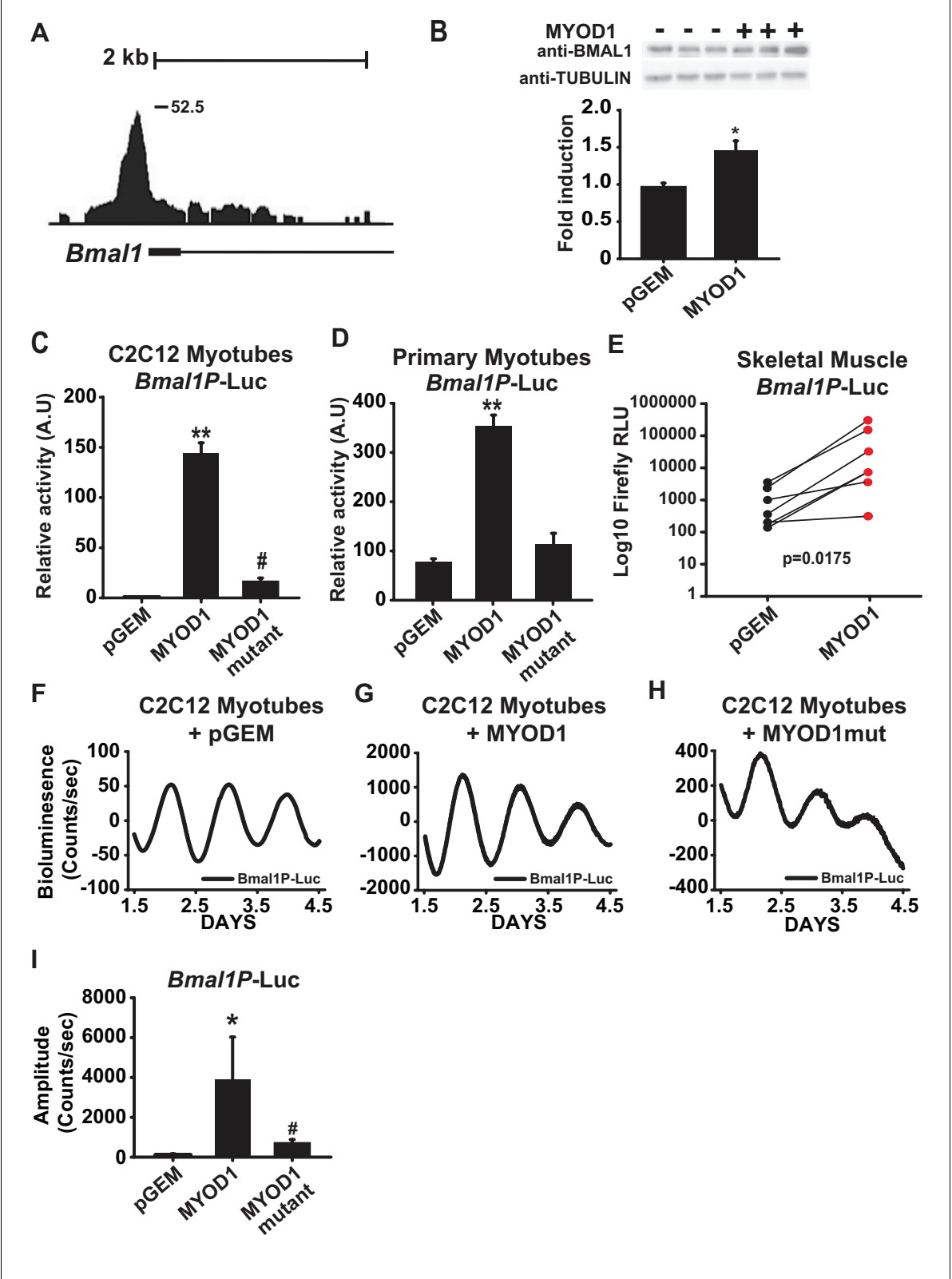

**Figure 2.** MYOD1 transactivates the *Bmal1* promoter and enhances its circadian amplitude in C2C12 myotubes. (**A**) UCSC genome browser visualization (mm10 genome) of MYOD1 binding tags within the *Bmal1* locus. (**B**) Representative BMAL1 western blots from C2C12 myoblasts transiently transfected with 150 ng pGEM empty vector (-) or MYOD1 expression vector (+). Densitometric values are expressed as average fold-change of BMAL1 over the Tubulin loading control which was unchanged with MYOD1 transfection ± SEM (n = 3). Results were analyzed with one-way ANOVA,

*Figure 2 continued on next page*

*Figure 2 continued*

* indicates a p-value less than 0.05 (p = 0.025). *Bmal1P*-Luc luminescence in C2C12 myotubes (2**C**, n = 4 biological replicates) and skeletal muscle primary myotubes (**D**, n = 3 biological replicates) with transient transfection of pGEM control, MYOD1, or MYOD1mut expression vectors. Luciferase activity for each co-transfection is plotted as average fold-change in relation to the pGEM empty vector control ± SEM. Results were analyzed using one-way ANOVA, ** indicates a p-value less than 0.001. # indicates a p-value less than 0.01 comparing MYOD1 vs MYOD1mut. (**E**) *Bmal1P*-Luc luminescence in electroporated skeletal muscle. *Bmal1P*-Luc activity was normalized to *Renilla* luciferase as an electroporation control with the right-leg receiving MYOD1 expression vector (red circles) and the left leg receiving the pGEM empty vector control (black circles). The p-value statistic was calculated by performing a Mann-Whitney non-parametric t-test. Note, one outlier was removed from each group based upon the Robust regression and outlier removal (ROUT) test, with a false discovery rate of <0.01. *Bmal1P*-Luc Dual-Luciferase activities in skeletal muscle primary myotubes (**F-H**) Representative *Bmal1P*-Luc driven bioluminescence recordings in synchronized C2C12 myotubes co-transfected with pGEM control (F, n = 7), MYOD1 (G, n = 6), or MYOD1mut (H, n = 4) expression vectors. Luminescence recordings are expressed as average counts/sec (base-line subtracted) (I) Average *Bmal1P*-Luc amplitudes ± SEM calculated by JTK_CYCLE from 1.5 to 4.5 days post-synchronization. Results were analyzed with one-way ANOVA, ** indicates p-value less than 0.001, # indicates p-value less than 0.002 (n = 4 biological replicates per group).

DOI: https://doi.org/10.7554/eLife.43017.003

enhance *Bmal1P*-Luc bioluminescence in comparison to the pGEM control and the magnitude of this induction was similar to what we observed in C2C12 myotubes (*Figure 2E*).

We next asked if MYOD1 influences the temporal or circadian oscillatory parameters of *Bmal1* expression with real-time bioluminescence recording of synchronized C2C12 myotubes transfected with the *Bmal1P*-Luc reporter in a Lumicycle. We found that *Bmal1P*-Luc displayed a robust circadian bioluminescence activity (*Figure 2F*). Consistent with the data in *Figure 2A–E*, over-expression of MYOD1 significantly enhanced the amplitude of the *Bmal1P*-Luc oscillation by approximately 40-fold in C2C12 myotubes (*Figure 2G,I*) indicating a novel role for MYOD1 as a modulator of core clock amplitude in skeletal muscle. In comparison to the wildtype MYOD1 vector, co-expression with the MYOD1mut vector failed to fully transactivate *Bmal1P*-Luc (*Figure 2H*), and only increased the amplitude of *Bmal1P*-Luc on average ~8 fold in comparison to the pGEM control (*Figure 2H,I*).

## MYOD1 regulates *Bmal1* via a non-canonical E-box element

The presence of a MYOD1 binding peak within the *Bmal1* locus (*Figure 2A*, *Table 1*), together with MYOD1's ability to transcriptionally activate and enhance the amplitude of the *Bmal1P*-Luc reporter argued for the presence of a MYOD1-response element within the *Bmal1* promoter. The *Bmal1P*-Luc construct contains 394 bp upstream of transcription start (TSS) and 154 bp downstream of the TSS, and preliminary truncations experiments indicated that the region downstream of the TSS was required for MYOD1-mediated transactivation (data not shown). To identify the possible response element, we performed three targeted truncations within the downstream region of the *Bmal1P*-Luc reporter (*Figure 3A*): T1 (from −394 to +86), T2 (from −394 to +68) and T3 (−394 to +33). Compared with the full length *Bmal1P*-Luc reporter, the T2 and T3 truncated reporters displayed significantly decreased response to MYOD1 over-expression (*Figure 3B*). Interestingly, we found a non-canonical E-box motif (5'-CAGGGA-3') within the +68 to+86 region. To determine if this site was required for the MYOD1 transactivation we performed site-directed mutagenesis (5-CAGGGA-3' to 5'-ATCTAA-3' , *Bmal1Pmut-Luc*) within the full-length reporter. The *Bmal1Pmut*-Luc reporter displayed a significantly blunted response to wildtype MYOD1 over-expression compared to the full length *Bmal1P-Luc* reporter (40 vs. 150-fold) (*Figure 3B*). We did note, however, that over-expression of MYOD1 was still sufficient to activate the *Bmal1Pmut*-luc reporter compared to the pGEM negative control and the magnitude of the transactivation was similar to that observed in the T2 and T3 truncation mutants. These findings argue that there is an additional cis element(s) upstream of +33 within the *Bmal1* promoter that is either directly or indirectly responsive to MYOD1 expression.

We next tested whether the MYOD1 response element identified above is required for the MYOD1-mediated enhancements to the *Bmal1P*-Luc circadian amplitude. As seen in *Figure 3C*, *Bmal1Pmut*-Luc displayed a ~ 24 hr circadian oscillatory pattern, but the amplitude of the oscillation was on average 50% lower than that of the wildtype *Bmal1P*-Luc reporter at basal conditions. Over-expression of MYOD1 was still able to significantly increase the circadian amplitude of the *Bmal1P-mut*-luc reporter (*Figure 3D,E*), however the amplitude of the *Bmal1Pmut*-Luc was only 10% of what we observed for *Bmal1P*-Luc (see *Figure 2I*). To extend the C2C12 results cells we also tested this

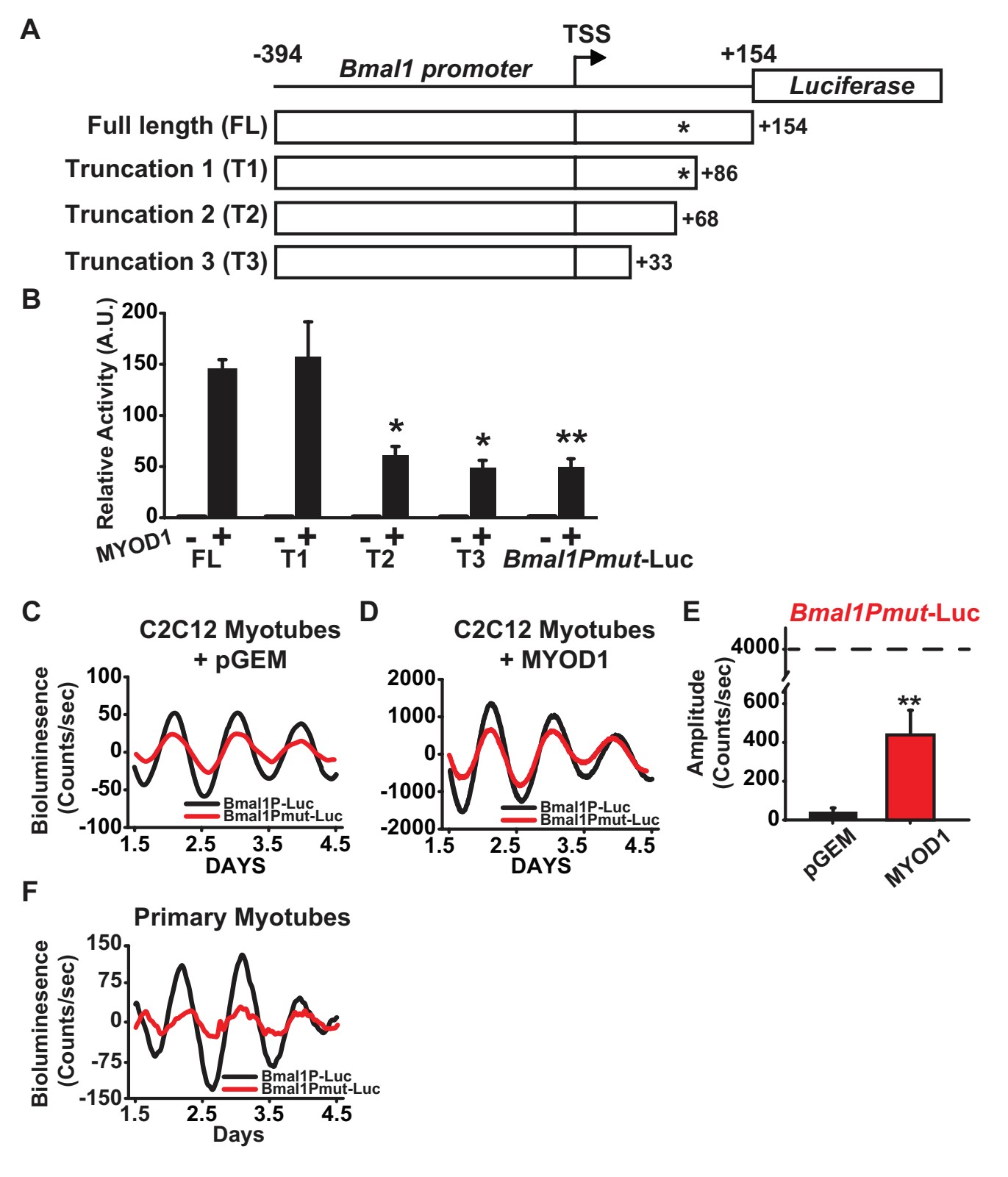

**Figure 3.** A non-canonical E-box within the *Bmal1* promoter is required for amplification by MYOD1. (**A**) Representative diagram of the *Bmal1P*-Luc promoter truncation constructs. TSS indicates transcriptional start site. * indicates the relative location of the non-canonical E-box element. FL indicates the full-length *Bmal1P*-Luc reporter. T1, T2 and T3 indicate the truncated reporters. (**B**) Dual-luciferase activities from the full-length *Bmal1P*-Luc, T1-T3, and *Bmal1Pmut*-Luc reporters (n = 4/group). Luciferase activity for each co-transfection is plotted as average fold-change in relation to the pGEM

*Figure 3 continued on next page*

*Figure 3 continued*

empty vector control ±SEM (n = 4). Results were analyzed using one-way ANOVA, ** indicates a p-value less than 0.001 in relation to the FL *Bmal1P*-Luc + MYOD1 result (p value < 0.001), * indicates a p-value less than 0.029 in relation to the FL *BmalP*-Luc +MYOD1 result. (C-D) Representative *Bmal1P*-Luc (black) and *Bmal1Pmut*-Luc (red) driven bioluminescence recordings in synchronized C2C12 myotubes co-transfected with the pGEM control vector (C), or MYOD1 expression vector (D). Luminescence recordings are expressed as average counts/sec (base-line subtracted). (E) Average *Bmal1Pmut*-Luc amplitudes calculated by JTK_CYCLE from 1.5 to 4.5 days post-synchronization (n = 3/group). Data are displayed as average amplitude ± SEM . Results were analyzed with one-way ANOVA, ** indicates p-value less than 0.001. (F) Representative *Bmal1P*-Luc (black) and *Bmal1Pmut*-Luc (red) driven bioluminescence recordings in synchronized skeletal muscle primary myotubes. Luminescence recordings are expressed as average counts/sec (base-line subtracted) for n = 3 biological replicates per group.

DOI: https://doi.org/10.7554/eLife.43017.005

region in primary myotubes and found that *Bmal1Pmut*-Luc displayed a dampened circadian amplitude compared to the wildtype *Bmal1P*-Luc reporter (*Figure 3F*). Taken together, these findings argue that the non-canonical E-box element located downstream of the *Bmal1* TSS is a MYOD1-response element and is required for full transactivation of the *Bmal1* promoter by MYOD1.

## MYOD1 co-localizes with both BMAL1 and CLOCK within myonuclei

The core molecular clock components are ubiquitously expressed in most, if not all cells, however the circadian transcriptomes in different tissues are highly divergent with approximately 5–10% overlap (*Zhang et al., 2014*; *Mure et al., 2018*). The mechanisms that direct tissue-specific circadian gene signatures are not fully understood; however, interactions between lineage specific factors and the core clock components have previously been reported (*Dufour et al., 2011*; *Lee et al., 2012*; *Perelis et al., 2015*; *Peek et al., 2017*; *Trott and Menet, 2018*). We reasoned that MYOD1 is a likely candidate for generating skeletal muscle transcriptional rhythms given that it oscillates in a robust fashion and is a master regulator of the myogenic gene program (*Andrews et al., 2010*). Therefore, we asked whether MYOD1 and BMAL1:CLOCK co-localize within nuclei of the C2C12 muscle cell line by performing a bimolecular fluorescence complementation (BiFC) assay.

We used a *Venus* based system with constructs tagged with either non-fluorescent *Venus* C- terminal fragments (VC) or with the non-fluorescent *Venus* N-terminal fragment (VN). With this design, *Venus* fluorescence signals are generated only when VC- and VN-tagged proteins co-localize and bring the two domains into physical association. We observed fluorescence signals within myonuclei when we co-expressed the known binding partners VN-HDAC5 with VC-MEF2c which served as a positive control and demonstrated that the BiFC assay was functional in our hands (*Figure 4—figure supplement 1E*). In agreement with BMAL1 and CLOCK forming heterodimers, we observed nuclear fluorescent signals when we co-expressed VN-CLOCK with VC-BMAL1 (*Figure 4A*). We performed a series of negative control experiments by co-expressing VN-BMAL1 with the VC-control vector (*Figure 4—figure supplement 1F*), or VN-CLOCK with the VC-control vector (*Figure 4—figure supplement 1I*) and failed to observe fluorescent signals. As an additional negative control, we could not detect a fluorescent signal when we co-expressed VN-BMAL1 with VC-BMAL2 (*Figure 4B*), consistent with reports that these isoforms do not heterodimerize. Interestingly, we observed nuclear fluorescent signals when we co-expressed VC-MYOD1 with VN-CLOCK (*Figure 4C*) or VN-BMAL1 (*Figure 4D*), indicating that MYOD1 co-localizes with both BMAL and CLOCK within myonuclei. We also observed fluorescent signals when we co-express VC-CLOCK or VC-BMAL1 with VN-MYOD1 (data not shown). It was interesting to note that MYOD1 was localized with either BMAL1 or CLOCK in multiple foci within the myonucleus indicating the role of MYOD1 and the clock factors across several genomic loci. However, the pattern of MYOD1:CLOCK factor binding was more discrete when compared to the diffuse binding of BMAL:CLOCK throughout the nucleus. This suggests that MYOD1 is interacting with only a subset of the BMAL1:CLOCK heterodimers within the nucleus. However, these images provide one snapshot in time and these transcription factors have a temporal component to their function that cannot be extrapolated from these images. Future studies will focus on the temporal interactions between MYOD1 and the clock factors as well as the fundamental molecular links and likely co-factors that modulate these interactions.

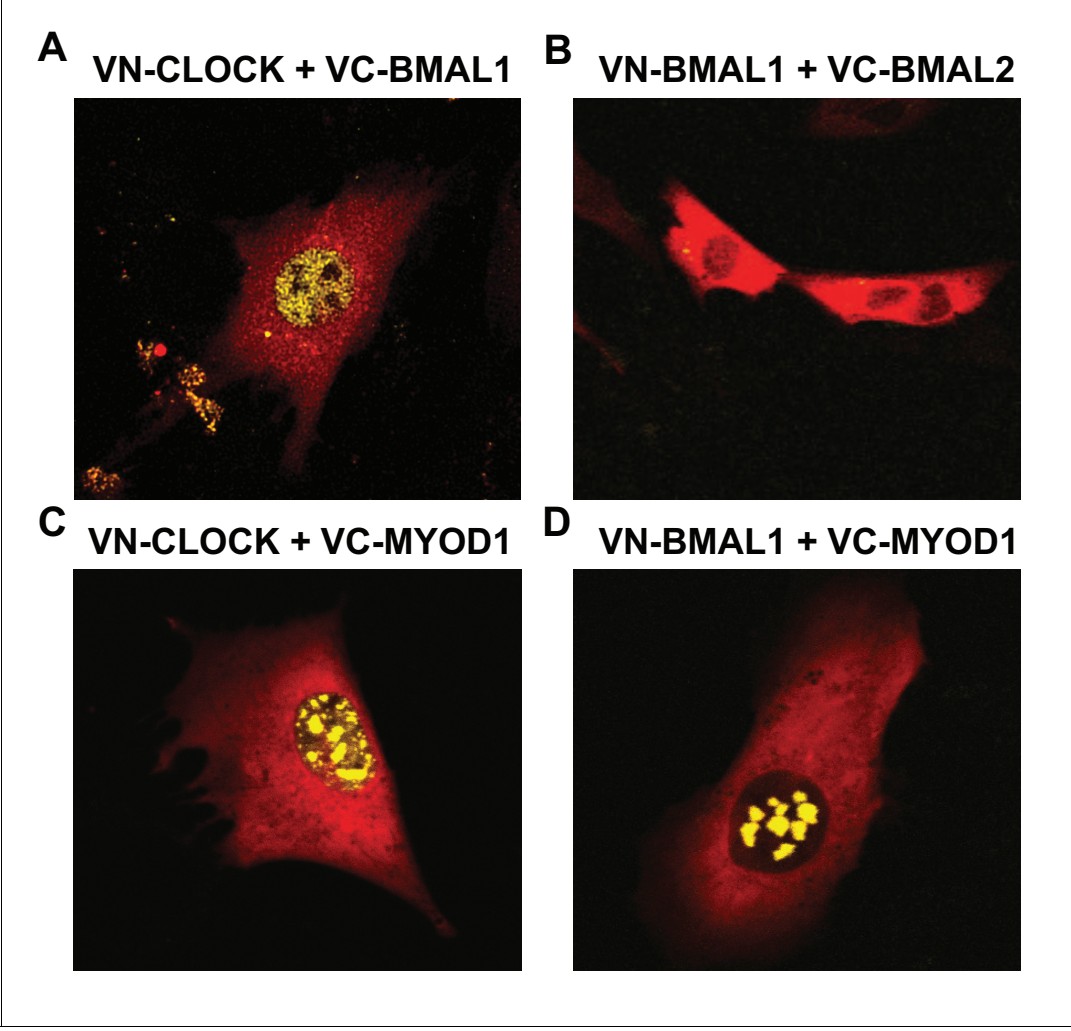

**Figure 4.** MYOD1 colocalizes with BMAL1 and CLOCK in C2C12 myonuclei. Representative images from the BiFC assay performed in C2C12 myoblasts co-transfected with (A) VN-CLOCK and VC-BMAL1, (B) VN-BMAL1 and VC-BMAL2, (C) VN-CLOCK and VC-MYOD1, and (D) VN-BMAL1 and VC-MYOD1. mCherry expression plasmids (red fluorescence signal) were co-transfected in each experiment to visualize the myoblasts and ensure successful transfection. Yellow fluorescence signals indicate positive co-localization via the formation of the *Venus* Luciferase.
DOI: https://doi.org/10.7554/eLife.43017.006

The following figure supplement is available for figure 4:

**Figure supplement 1.** Representative control images for BiFC experiments in C2C12 myotubes.
DOI: https://doi.org/10.7554/eLife.43017.007

### Selection of *Titin-cap* (*Tcap*) as a model gene for studying regulatory interactions between MYOD1 and BMAL1:CLOCK in skeletal muscle

Since MYOD1 co-localized with BMAL1:CLOCK, we next wanted to address if MYOD1 works with these core clock factors to regulate the circadian expression of muscle genes. To study the potential transcriptional mechanisms between MYOD1 and BMAL1:CLOCK we first performed a bioinformatics filtering paradigm to identify a model muscle-specific gene to test MYOD1 and BMAL1:CLOCK interactions (*Supplementary file 5*). We started our analysis with genes that displayed robust rhythmic expression in skeletal muscle (*Hughes et al., 2010*; *Zhang et al., 2014*) by setting a JTK_CYCLE circadian p-value cutoff of <0.001. From this list we selected genes that reach peak gene expression at the inactive to active phase transition (peak between CT 10–14) similar to known circadian genes *Dbp* or *Per2*, as we reasoned that those genes are more likely to be direct BMAL1:CLOCK targets. We further filtered our list by selected genes that are significantly down-regulated in the inducible

skeletal muscle-specific *Bmal1* knockout model (iMS-*Bmal1*^-/-^) (*Hodge et al., 2015*). From this analysis we selected the z-line protein *Titin-cap* (*Tcap,* also known as *Telethonin*). Given that previous reports have reported *Tcap* regulation by both MYOD1 in vitro (*Zhang et al., 2011*) and BMAL1 in the heart (*Podobed et al., 2014*), we reasoned that it is a good candidate for testing transcriptional regulatory interactions between MYOD1 and BMAL1:CLOCK.

## *Tcap* is synergistically activated by MYOD1 and BMAL1:CLOCK

To determine if *Tcap* is regulated by BMAL1:CLOCK in skeletal muscle we performed Dual-Luciferase reporter assays in C2C12 myotubes by transfecting BMAL1 and CLOCK expression vectors with a *Tcap* promoter-Luciferase reporter (*TcapP*-Luc). We found that co-expression of BMAL1 and CLOCK (BMAL1:CLOCK) enhanced *TcapP*-Luc activity approximately 2.5-fold compared to the pGEM control (*Figure 5A*). Over-expression of mutant forms of either BMAL1 (BMAL1mut) or CLOCK (CLOCKmut) failed to transactivate the *TcapP*-Luc reporter suggesting that a functional BMAL1:CLOCK heterodimer is necessary to activate the *Tcap* promoter (*Figure 5A*). We next performed an siRNA mediated knockdown of *Bmal1* to compliment our *BMAL1mut* findings and determine if expression of endogenous *Tcap* levels are dependent on BMAL1. We found that *Bmal1* knockdown significantly reduced the expression of *Tcap* as well as *MyoD1* mRNAs in C2C12 myotubes in comparison to a non-targeting control siRNA (*Figure 5—figure supplement 1*). These findings are in agreement with observed loss of *Tcap*'s circadian expression in iMS-*Bmal1*^-/-^ mouse models (*Supplementary file 5*) and further demonstrate that endogenous *Tcap* expression is regulated downstream of BMAL1.

Given the observation that MYOD1 co-localized with BMAL1 and CLOCK within myonuclei, we next sought to determine whether *Tcap* is cooperatively regulated by BMAL1:CLOCK and MYOD1. Over-expression of MYOD1 resulted in a robust activation of *TcapP*-Luc (~280 fold), while over-expression of the MYOD1mut vector failed to activate the reporter to the same degree (only 7-fold) (*Figure 5B*). Over-expression of MYOD1 together with BMAL1:CLOCK resulted in a synergistic activation of the *TcapP*-Luc reporter (~540 fold) (*Figure 5B*), that was significantly greater than MYOD1 alone (p-value = 0.033). The cooperative interaction among these factors is further highlighted by the reduced ability of MYOD1 to transactivate *TcapP*-Luc when co-expressed with mutant forms of either BMAL1 or CLOCK. Over-expression of MYOD1 with BMAL1mut or CLOCKmut reduced the *TcapP*-Luc activity from ~540 fold to ~55 fold (BMAL1mut) and ~180 fold (CLOCKmut) (*Figure 5B*, p-value < 0.01). Taken together, these results argue that BMAL1:CLOCK and MYOD1 work in a cooperative fashion to regulate *Tcap* expression.

## MYOD1 enhances the circadian amplitude of *Tcap*

The *TcapP*-Luc reporter displayed a robust circadian oscillation in synchronized C2C12 myotubes (*Figure 5C*), and, as predicted, in an anti-phasic expression pattern compared to *Bmal1P*-Luc (as seen in *Figure 2F*). These findings validate that the *TcapP*-Luc reporter contains the necessary regulatory elements required for its circadian oscillation in C2C12 cells. Given that MYOD1 and BMAL1:CLOCK synergistically activated the *TcapP*-Luc reporter in our Dual Luciferase assays, we next tested whether over-expression of MYOD1 influences the circadian oscillatory parameters of *Tcap*. Over-expression of MYOD1 alone significantly elevated the circadian amplitude of *TcapP*-Luc over 20-fold (*Figure 5D,F*). Interestingly, over-expression of the MYOD1mut vector failed to amplify the *TcapP*-Luc rhythm to the same extent as the wildtype MYOD1 vector (*Figure 5E,F*). To determine if MYOD1 influences the circadian expression profile of endogenous *Tcap* mRNA, we quantified *Tcap* expression over a circadian time-course in MYOD1-CE and wildtype control muscle tissue. *Tcap* mRNA displayed a dampend expression (*Figure 5G*) and complete loss of circadian rhythmicity in MYOD1-CE muscle tissue in comparison to wiltype controls (*Figure 5G,H*). These findings are consistent with the results from the dual luciferase assays and demonstrate the requirement for both BMAL1 and MYOD1 in promoting circadian rhythm expression of *Tcap*. Additionally, the findings support the BiFC results that MYOD1 and the molecular clock factors function as a transcriptional regulatory complex in muscle.

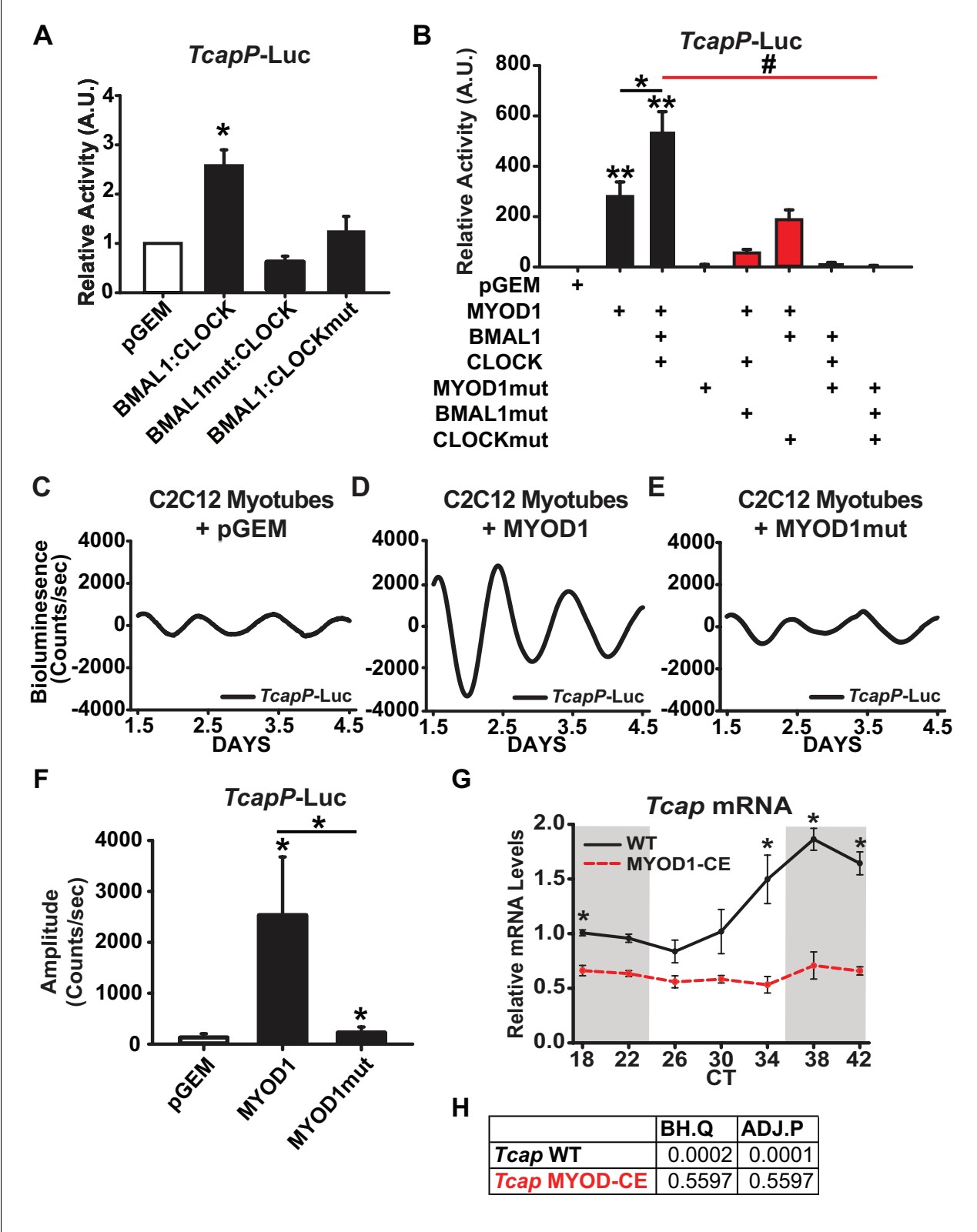

**Figure 5.** *TcapP*-Luc is synergistically activated by MYOD1 and BMAL1:CLOCK. (**A**) *TcapP*-Luc Dual-Luciferase activity responses from C2C12 myotubes co-transfected with BMAL1 +CLOCK, BMAL1mut + CLOCK, and BMAL1 +CLOCKmut vectors (n = 4 biological replicates/group). Luciferase activity for each co-transfection is plotted as average fold-change in relation to the pGEM empty vector control ± SEM. Results were analyzed using one-way ANOVA, * indicates p = 0.029. (**B**) *TcapP*-Luc Dual-Luciferase activity responses from C2C12 myotubes co-transfected with MYOD1 alone,
*Figure 5 continued on next page*

Figure 5 continued

MYOD1 +BMAL1+CLOCK, or variations of BMAL1mut, CLOCKmut, and MYOD1mut vectors. Luciferase activity for each co-transfection is plotted as average fold-change in relation to the pGEM empty vector control ± SEM (n = 4). Results from MYOD1 and MYOD1 +BMAL1:CLOCK co-transfections were analyzed using one-way ANOVA, ** indicates a p-value less than 0.001 in relation to the pGEM control. Results from the mutant co-transfection experiments (red) were analyzed using one-way ANOVA. # indicates a p-value less than 0.01 relative to the MYOD1 +BMAL1:CLOCK result. (C–E) Representative *TcapP*-Luc driven bioluminescence recordings in synchronized C2C12 myotubes co-transfected with the pGEM control (C), MYOD1 (D), or MYOD1mut (E) expression vectors. Luminescence recordings are expressed as average counts/sec (base-line subtracted). (F) Average *TcapPmut*-Luc amplitudes calculated by JTK_CYCLE from 1.5 to 4.5 days post-synchronization. Data are displayed as average amplitude ± SEM (n = 3 biological replicates/group). Results were analyzed with one-way ANOVA, * indicates p-value less than 0.05. (G) *Tcap* temporal mRNA expression profiles from muscles of MYOD1-CE (dotted red) or C57BL/6J (solid black) mice. Dark shading indicates relative dark/active phase as these mice were reared in DD at the time of collection. At each time-point RT-PCR expression values are displayed as average fold-change relative to the *Rpl26* house-keeping gene ± SEM (n = 3). Results were analyzed with one-way ANOVA comparing WT vs. MYOD1-CE at each time-point, * indicates a p-value less than 0.05. (H) JTK_CYCLE statistics for the RT-PCR results corresponding to *Tcap*'s temporal expression values. 'BH.Q' column reports false discover rates and 'ADJ.P' reports adjusted p-values.

DOI: https://doi.org/10.7554/eLife.43017.008

The following figure supplement is available for figure 5:

**Figure supplement 1.** Effect of *Bmal1* siRNA mediated knockdown on *MyoD1* and *Tcap* expression.

DOI: https://doi.org/10.7554/eLife.43017.009

## A tandem E-box element within the *Tcap* promoter is required for its circadian expression and synergistic regulation by MYOD1 and BMAL1: CLOCK

We next sought to investigate the cis-regulatory elements within the *Tcap* promoter that are required for its circadian expression pattern and synergistic activation by MYOD1 and BMAL1: CLOCK. The *Tcap* promoter contains three canonical E-box motifs that are highly conserved in mammals (*Zhang et al., 2011*) which consist of a distal tandem E-box element (labelled E2-5' and E2-3') and a proximal single E-box (labelled E1) (*Figure 6A*). We performed site-directed mutagenesis at each of the three E-box elements within the *TcapP*-Luc reporter (*TcapP-E2-5'*-Luc, *TcapP-E2-3'*-Luc, and *TcapP-E1*-Luc) to investigate their individual role(s) in our Dual Luciferase and circadian bioluminescence reporter assays. We first evaluated the activity of these mutant reporters in response to BMAL1:CLOCK over-expression in C2C12 myotubes. BMAL1:CLOCK was still able to transactivate the E1 and E2-5' mutants, while the E2-3' mutant was unresponsive (*Figure 6B*). To our surprise, transactivation by MYOD1 was unaltered in all three of the *Tcap* mutants compared to the wildtype reporter (*Figure 6C*). However, mutation at any one of the three E-box elements completely abolished the synergistic activation by MYOD1 and BMAL1:CLOCK (*Figure 6C*) suggesting a complex interaction among these factors across these sites.

We next performed real-time bioluminescence recordings of the *TcapP*-Luc E-box mutants to evaluate whether they are required for *Tcap*'s circadian expression pattern. We observed a normal (~24 hr) circadian oscillatory pattern of *TcapP-E1*-Luc (*Figure 6D*), albeit with an ~2 fold dampening in the amplitude compared to the wildtype reporter. This is consistent with the dual luciferase results in *Figure 6C* that demonstate that E1 is required for full responsiveness to the synergistic regulation of MYOD1, BMAL1 and CLOCK. The real time bioluminescence assays did uncover that mutation of either the 3' or 5' tandem E-box motifs resulted in a complete loss of circadian oscillation (*Figure 6E,F*). In addition we found a similar pattern for *TapP*-Luc and the *TcapP-E2-5'*-Luc mutant reporter vectors in a primary myotubes (*Figure 6G*). These findings demonstrate that both E-boxes within the tandem E-box element are required for proper BMAL1:CLOCK-mediated regulation of *Tcap*'s circadian expression, while the E1 E-Box appears to be required for cooperative transactivation of *Tcap* by MYOD1 and BMAL1:CLOCK.

Our initial MYOD1 ChIP-Seq analysis did not yield a strong MYOD1 binding peak within the *Tcap* promoter region. Given our dual-luciferase and circadian lumicycle data we decided to test whether MYOD1 and BMAL1:CLOCK do indeed bind to the E-boxes within the *Tcap* promoter using a more targeted approach. We performed a MYOD1 and BMAL1 ChIP-PCR in adult skeletal muscle taken at ZT 2 ( two hours after lights on) with primers that incorporate either the *Tcap* tandem E-box or the E1 E-box. We found that BMAL1 was significantly enriched at both the E1 E-box (*Figure 7B*) as well as the tandem E-box element (*Figure 7D*). Interestingly, we found MYOD1 to be enriched within the

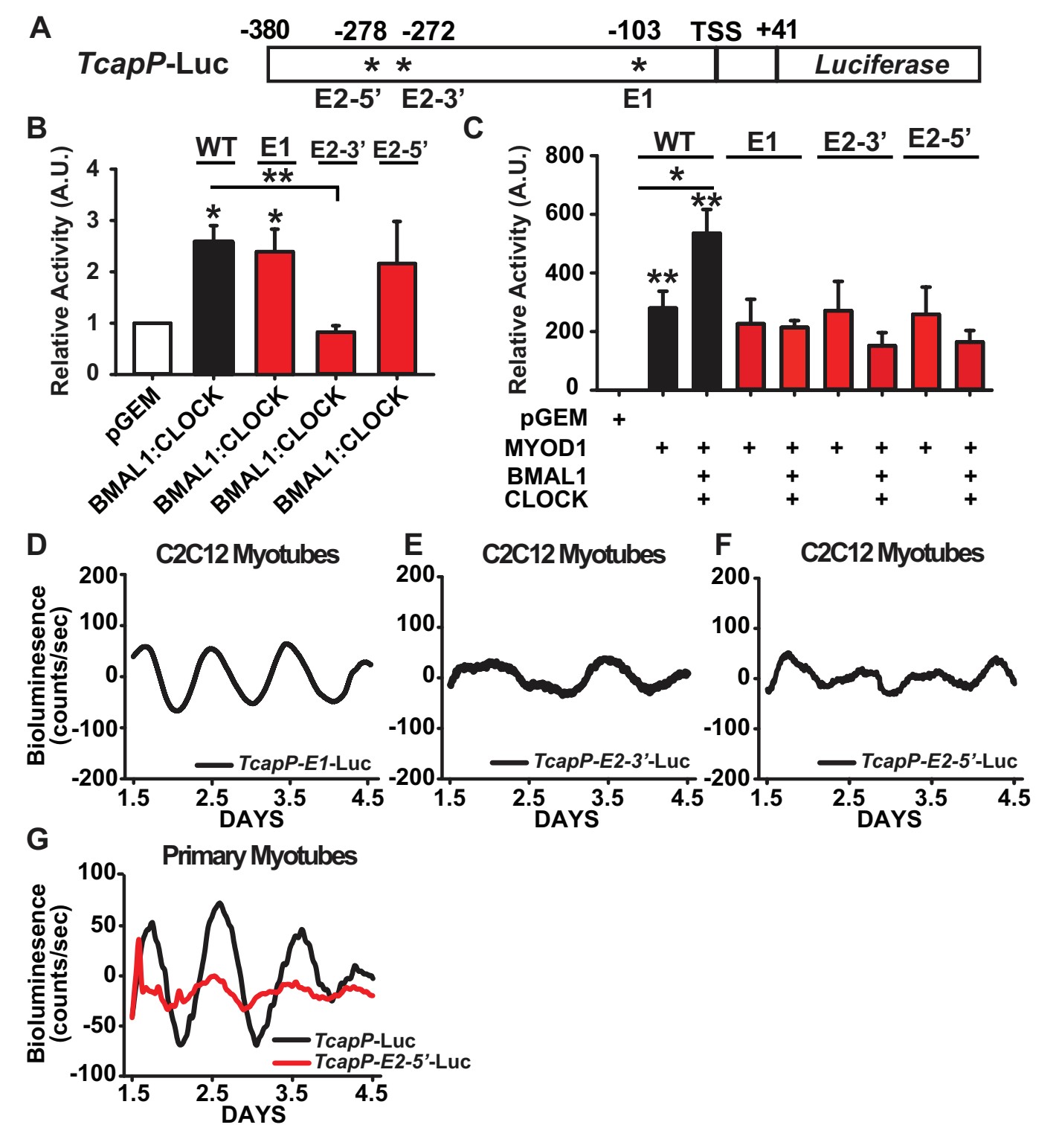

**Figure 6.** Synergistic activation of the *Tcap* promoter by MYOD1 +BMAL1:CLOCK requires the tandem E-box element. (**A**) Graphical representation of the *TcapP*-Luc promoter:reporter construct. TSS indicates transcript start site. E1 E-box is located at −103 and E2 tandem E box is located −272 (E2 3') and −278 (E2 5') from the TSS. (**B**) Dual-Luciferase activity responses from the wildtype *TcapP*-Luc and the *Tcap* E-box mutants in C2C12 myotubes co-transfected with BMAL1:CLOCK. Luciferase activity for each co-transfection is plotted as average fold-change in relation to the pGEM empty vector control ± SEM (E2-3' and E2-5' n = 4, E1 n = 3). Results were analyzed using one-way ANOVA, ** indicates p-value less than 0.01 in relation to the WT
*Figure 6 continued on next page*

*Figure 6 continued*

*TcapP*-Luc response, * indicates a p-value less than 0.05 in relation to the pGEM control vector. (C) Dual-Luciferase activity responses from the *Tcap* E-box mutants with co-transfection of MYOD1 alone or MYOD1 +BMAL1:CLOCK. Luciferase activity for each co-transfection is plotted as average fold-change in relation to the pGEM empty vector control ± SEM (n = 3). Results were analyzed using one-way ANOVA. In comparison to the pGEM control all co-transfections were significantly elevated (p-value < 0.05), ** indicates a p-value of less than 0.01 in relation to the pGEM control, and * indicates a p-value of less than 0.05 comparing *TcapP*-Luc with over-expression of MYOD1 vs MYOD1+BMAL1:CLOCK. In comparison to the pGEM control all co-transfections were significantly elevated (p-value<0.05). . No statistical differences were observed for each of the *Tcap* mutant reporters comparing MYOD1 alone to MYOD1 +BMAL1:CLOCK. (D-F) Representative bioluminescence recordings from the *TcapP-E1*-Luc (D), *TcapP-E2-3'*-Luc (E), and *TcapP-E2-5'mut*-Luc (F) in synchronized C2C12 myotubes. (G) Representative TcapP-Luc (black, n = 3 biological replicates) and TcapPmut-E2-5'-Luc (red, n = 3 biological replicates) driven bioluminescence recordings in synchronized skeletal muscle primary myotubes. Luminescence recordings are expressed as average counts/sec (base-line subtracted).

DOI: https://doi.org/10.7554/eLife.43017.010

tandem E-box (*Figure 7C*), albeit at a lower binding strength in comparison to BMAL1. We did not detect significant enrichment of MYOD1 binding to the E1 E-box element (*Figure 7A*). Taken together these results suggest that MYOD1 and BMAL1 regulate the circadian expression of *Tcap* via direct interactions at *Tcap*'s tandem E-box element but the weaker interaction with MYOD1 indicates that this binding may be mediated through a transcription factor complex and not direct DNA binding (*Figure 7E*).

## Discussion

Previous studies have demonstrated the vital importance of circadian time-keeping in skeletal muscle, however the molecular mechanisms responsible for maintaining robust clock function and circadian amplitude have yet to be identified (*Andrews et al., 2010*; *Chatterjee et al., 2013*; *Schroder et al., 2015*; *Harfmann et al., 2016*; *Liu et al., 2016*; *Schiaffino et al., 2016*). Herein we demonstrate that MYOD1 strengthens molecular clock amplitude in skeletal muscle by functioning as a direct transcriptional activator of *Bmal1*. We show that MYOD1 co-localizes with BMAL1:CLOCK in myonuclei and works in a synergistic fashion to drive the circadian expression of the muscle specific gene, *Tcap*. This study is the first to demonstrate functional roles for MYOD1 as an upstream, positive regulator of the core molecular clock, as well as a direct modulator of clock output in skeletal muscle.

Our lab previously reported that BMAL1:CLOCK rhythmically bind to the *MyoD1* promoter at the distal core-enhancer region resulting in its robust circadian expression in skeletal muscle (*Andrews et al., 2010*; *Zhang et al., 2012*). One of the main findings of our current study is that MYOD1 regulates circadian amplitude through direct transcriptional control of *Bmal1*, thus indicating that MYOD1 forms a transcriptional feed-forward loop with the molecular clock (*Figure 7E*). The temporal expression profile of MYOD1 correlates with its predictive role as an upstream regulator of *Bmal1*, as MYOD1 protein peaks in the mid-to-late active phase in adult skeletal muscle (*Andrews et al., 2010*) in anticipation of the upward rise in *Bmal1* expression which occurs during the early inactive phase. Our findings are also supported by a recent study that utilized an inducible MYOD1 expression system in human fibroblasts to define the genome architecture of human myogenic reprograming. *Liu et al. (2018)*, found that MYOD1 activation in fibroblasts was sufficient to synchronize core circadian genes, and the authors conclude that MYOD1 may serve to communicate exogenous entrainment cues to the clock (*Figure 7E*). Our results provide mechanistic insights to Liu et al's findings by demonstrating that MYOD1 may sychronize the molecular clock through direct transcriptional regulation of an enhancer within the core clock gene, *Bmal1*. This transcriptional feed-forward loop between MYOD1 and the clock shares mechanistic similarities with previously identified tissue-specific, peripheral modulators of clock amplitude which A) display diurnal variations in expression and B) regulate circadian amplitude via direct transcriptional control of the *Bmal1* promoter (*Onishi et al., 2008*; *Hirota et al., 2010*; *Wang et al., 2010*; *Poliandri et al., 2011*; *Valnegri et al., 2011*; *Lin et al., 2014*; *Wible et al., 2018*).

The accessory core clock transcription factors Reverbα and RORα are responsible for stabilizing period length and modulating circadian amplitude in a ubiquitous fashion across all tissues and cell-types (*Preitner et al., 2002*; *Guillaumond et al., 2005*; *Jetten, 2009*; *Solt et al., 2011*). Both

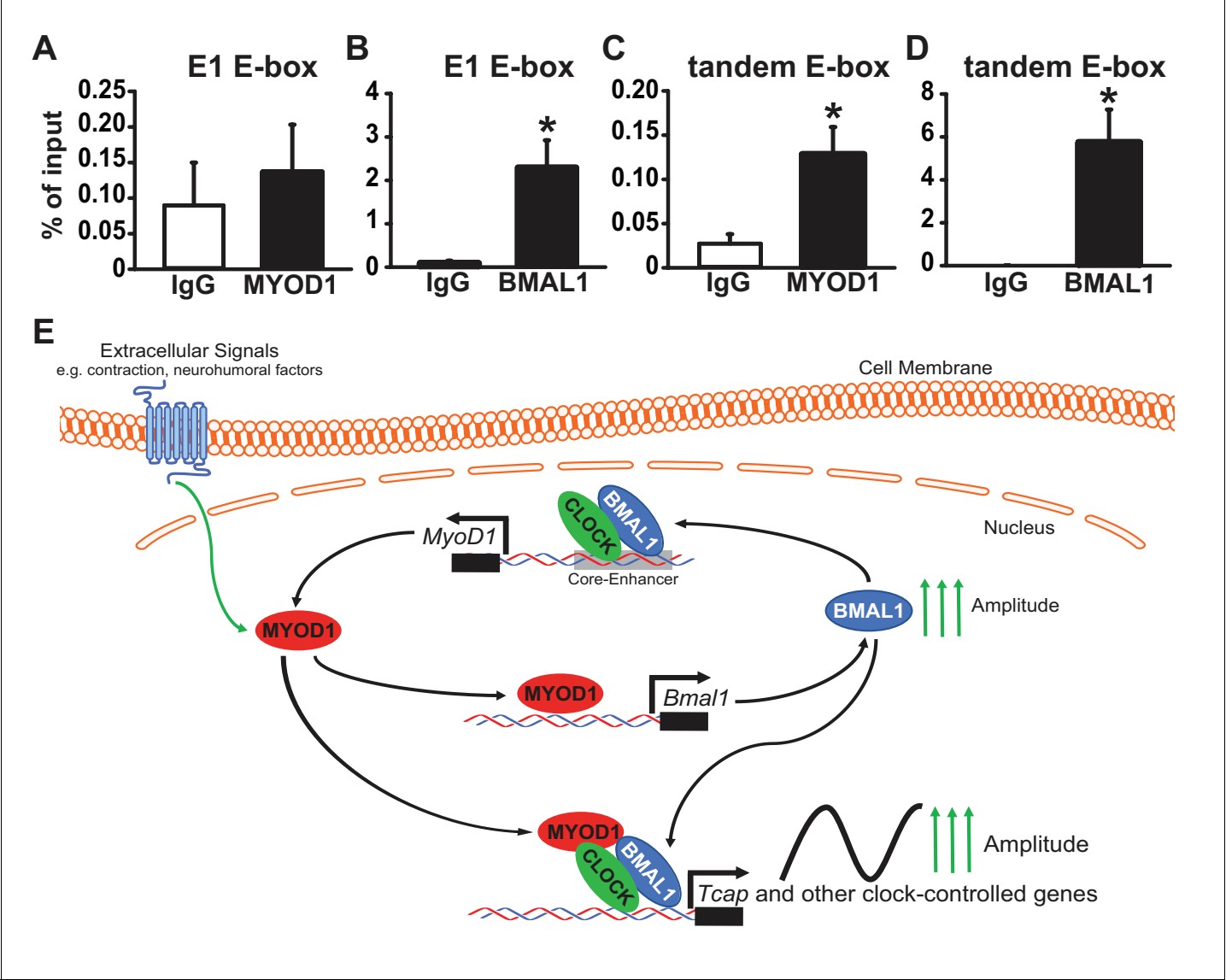

**Figure 7.** MYOD1 and BMAL1 bind to the *Tcap* tandem E-box element in adult skeletal muscle. (A-B) Chromatin Immunoprecipitation-PCR with anti-MYOD1 and -BMAL1 antibody pulldowns (and IgG controls) to detect binding of MYOD1 and BMAL1 within *Tcap*'s E1 E-box element (A, B) or the tandem E-box (C, D, primers contain 3' and 5' Eboxes). Pull-downs were performed with extracts from adult mouse quadricep muscles collected at ZT 2 (2 hours after lights on). Data are displayed as an average % of input ± SEM (n = 3 samples/group). Results were analyzed with a one-way ANOVA and * denotes a p-value ranging from 0.02 to 0.037. (E) Graphical model of depicting the role of MYOD1 in modulating core clock gene expression and working as a co-factor with BMAL1:CLOCK to amplify downstream circadian genes in skeletal muscle. MYOD1 activity is modulated by extracellular signals and it amplifies *Bmal1* expression via direct transcriptional activation. BMAL1:CLOCK in turn form a positive feedback loop to regulated the circadian expression of *MyoD1* by targeting the core-enhancer element. MYOD1 and BMAL1:CLOCK work in a synergistic fashion to amplify the expression of circadian genes.

DOI: https://doi.org/10.7554/eLife.43017.011

Reverbα and RORα are direct BMAL1:CLOCK target genes that compete for binding at the Rev-Erb Response element (RORE) within the *Bmal1* promoter thus forming a feedback loop where Rev-Erbα represses while RORα activates the expression of *Bmal1*. Interestingly, a number of peripheral modulators of circadian amplitude regulate *Bmal1* expression through RORα and/or REV-ERBα dependent mechanisms. For example, in liver tissue *Nrip1* promotes circadian amplitude through interacting with RORα at the *Bmal1* promoter (*Poliandri et al., 2011*). Conversely, the glucose responsive factor *Tieg1* decreases circadian amplitude by repressing *Bmal1* expression in a REV-

ERBα dependent fashion (*Hirota et al., 2010*; *Poliandri et al., 2011*). Interestingly, RORα has been shown to physically interact with MYOD1 at its N-terminal activation domain to promote myogenesis (*Lau et al., 1999*). Although we did not test this potential interaction in our current study, it is possible that MYOD1 regulates *Bmal1* in a RORα-dependent fashion which may help explain why the MYOD1mut vector, which lacks the transactivation domain, failed to activate the *Bmal1P*-Luc reporter (*Figure 2D,E,H*) to the same degree as the wildtype MYOD1 vector. Future studies will be aimed at investigating whether MYOD1 works alongside the accessory limb factors, Rev-erbα and/or RORα, to regulate *Bmal1* expression and circadian amplitude.

To investigate the potential mechanistic interactions between MYOD1 and BMAL1:CLOCK we decided to characterize their roles (both individually and together) in regulating a muscle specific circadian gene. Our bioinformatics approach led us to identify *Tcap* as a model gene for studying the interaction between MYOD1 and BMAL1:CLOCK. The novelty of these studies lie in the observations that MYOD1 and BMAL1:CLOCK work together to synergistically regulate the *Tcap* promoter. The transcriptional regulation of *Tcap* by MYOD1 appears to require a functional clock, as MYOD1-mediated activation of *TcapP*-Luc was dampened with over-expression of mutant forms of BMAL1 or CLOCK. Taken together with our findings that MYOD1 co-localizes with BMAL1:CLOCK and targets ~ 30% of the skeletal muscle circadian transcriptome, we propose a model by which MYOD1 regulates downstream circadian genes through forming cooperative interactions with BMAL1:CLOCK at cis-regulatory regions. In support of this model, a recent BMAL1 ChIP-Seq in mouse skeletal muscle revealed an enrichment for the canonical MYOD1 motif (5'-CAGCTG-3') within BMAL1 binding sites (*Dyar et al., 2018*).

Although the canonical BMAL1:CLOCK E-box binding sequence is CACGTG, recent studies have identified the presence of tandem E-box elements (2 E-boxes separated by six nucleotides) within a majority of core clock genes (*Nakahata et al., 2008*; *Tokuda et al., 2017*). Our functional characterization *Tcap*'s tandem E-box site provides a deeper understanding into the mechanisms of circadian gene regulation by MYOD1 and the clock factors. We found that the tandem E-box within the *Tcap* promoter is directly targeted by MYOD1 and BMAL1 in adult skeletal muscle and is functionally required for *Tcap*'s circadian regulation and synergistic activation by MYOD1 and BMAL1:CLOCK. In support of the BMAL1 binding enrichment within *Tcap*'s tandem E-box, we observed a loss in the circadian oscillation with either the 3' or 5' *TcapP*-Luc mutants. Furthermore, the 3', but not the 5', E-box mutant was unresponsive to BMAL1:CLOCK over-expression indicating a potential preference for BMAL1:CLOCK binding at the distal E-box. The observation that all three E-box mutants were responsive to MYOD1 over-expression, but were arrhythmic argues that *Tcap*'s circadian expression pattern is set primarily by BMAL1:CLOCK and not by MYOD1. This interpretation is in agreement with previous reports that *Tcap* oscillates and is bound by BMAL1 at the tandem E-box in heart tissue, where MYOD1 expression/activity is non-existent (*Podobed et al., 2014*). Taken together these findings argue that *Tcap*'s rhythmic oscillation is set primarily by BMAL1:CLOCK, while MYOD1 functions as a peripheral rheostat that controls the amplitude (i.e. gain) of the core clock mechanism and downstream clock-driven transcripts in skeletal muscle.

MYOD1's regulation of circadian amplitude may have evolved as a means for communicating muscle specific time-cues to the clock. It is well established that molecular clocks sense and respond to environmental time-cues in a tissue-specific fashion (i.e. light vs. nutrient entrainment) (*Podobed et al., 2014*; *Tokuda et al., 2017*). For instance, the intrinsically photosensitive retinal ganglion cells signal light information to the SCN, while peripheral clocks, which are incapable of sensing light directly, are more responsive to feeding cues (*Podobed et al., 2014*). However, it is not fully understood how clocks in different tissues sense specific time-cues, but recent studies have begun to identify factors that communicate environmental signals to the clock. For instance, Oligophrenin-1 which is activated by synaptic firing, modulates the expression of *Bmal1* by sequestering REV-ERBα within the cytosol of hippocampal neurons (*Poliandri et al., 2011*). Additionally, Hif1α which is activated during hypoxic conditions has been shown to influence clock gene expression within a number of cell types including skeletal muscle (*Trott and Menet, 2018*). Ourselves and others have demonstrated that muscle contractions and nutrient cues serve as time cues for the clock mechnism within skeletal muscle, as forced exercise or meal timing phase shifts core clock rhythms (*Wolff and Esser, 2012*; *Shavlakadze et al., 2013*; *Nakao et al., 2015*; *Liu et al., 2016*). MYOD1 is a likely candidate for communicating these environmental time-cues to the clock, as its expression and activity is strongly influenced by nutrient signals (repressed during fasting conditions)

and muscle contraction (elevated with denervated muscle tissue), both of which are strong time-cues for the skeletal muscle molecular clock (*Figure 7E*) (*Wolff and Esser, 2012*; *Shavlakadze et al., 2013*). MYOD1 activity may also mediate information for the timing or phase of the clock rhythms as we observed phase delays in *Bmal1P*-Luc rhythms with over-expression of MYOD1 in C2C12 myotubes.

In summary, we demonstrate that MYOD1 is a critical transcription factor in the regulation of circadian amplitude and downstream rhythmic gene expression in skeletal muscle. Given the vital role clocks play in the daily maintenance of skeletal muscle physiology and metabolism (*Andrews et al., 2010*; *Chatterjee et al., 2013*; *Schroder et al., 2015*; *Harfmann et al., 2016*; *Liu et al., 2016*; *Schiaffino et al., 2016*), these findings provide potential therapeutic avenues for enhancing clock function in skeletal muscle through targeting MYOD1. The synergistic interactions observed between MYOD1 and BMAL1:CLOCK in the circadian regulation of *Tcap* may also serve as a basis for understanding how core clock components interact with tissue-specific transcriptional networks.

# Materials and methods

## Animals

All the animal procedures in this study were conducted in accordance with the guidelines of University of Florida for the care and use of laboratory animals. C57BL/6J male mice were purchased from Jackson Laboratory and the MYOD-CE mice, initially described by *Chen and Goldhamer (2004)*, are maintained in a breeding colony within the Animal Care Services at the University of Florida.

## Circadian tissue collection and RT-PCR

We followed the established protocol in the circadian field for collection of tissues for circadian analysis (*Zhang et al., 2012*). Briefly, wildtype and MYOD1-CE mice (10–12 weeks of age) were entrained to a 12 hr light/12 hr dark (12L:12D) cycle for two weeks and then placed in total darkness for 30 hr prior to the start of the tissue collection to remove any light cues. Tissues were collected in darkness with red light from mice (n = 3/genotype) every 4 hr for 28 hr (seven time points). By circadian convention, the onset of activity for a nocturnal animal is defined as circadian time 12 (CT12) so the first collection was performed at CT18. Total RNA was extracted using Trizol Reagent (Ambion) and purified using the RNeasy mini kit (Qiagen, Cat. 74106). 1000 ng of total RNA was reverse transcribed using SuperScript III first strand cDNA synthesis system (Invitrogen, 18080–051). RT-PCR was performed on a QuantiStudio three thermal cycler (Applied Biosystems) from 20 ng of cDNA in a total volume of 20 µl containing Fast SYBR Green Master Mix (Applied Biosystems, Cat. 4385612) and 400 nM of each primer (*Supplementary file 6*). mRNA levels of target genes were normalized to corresponding *Rpl26* mRNA levels and relative quantification was calculated by using the standard curve method. Circadian statistics were performed with JTK_CYCLE analysis (*Hughes et al., 2010*) R-package to determine circadian p-values.

## Targeted ChIP-PCR in adult skeletal muscle

Wildtype (C57BL/6J, 10–12 weeks of age) mice were entrained to a 12 hr light:dark cycle (12L:12D) for 2 weeks. Quadricep muscles were collected from two mice (approximately 600–700 mg) at ZT2, formaldehyde cross-linked, and homogenized. Isolated chromatin was then sonicated and purified as described in ChIP-Seq methods. 20 µl of supernatant was used as input control. The remaining supernatant was split into three equal parts and incubated over-night at 4°C with antibodies for BMAL1, MYOD1, or rabbit IgG. Immune-complexes were then incubated with 20 µl protein Dynabeads A/G beads (Invitorgen 10003D, 10001D) for 2 hr. The beads were then washed with 1 ml RIPA buffer 3-times and 1 ml TE buffer once with a magnetic stand (each wash lasted approximately 5 min). DNA was eluted with 150 µl of 20 mM Tris pH 7.5, 5 mM EDTA, 0.5% SDS at 65°C for 30 min using a thermal mixer. Magnets were used to separate the beads from the supernatant, which was then transferred into fresh tubes with 6 µl of 5 M NaCl and 2 µl of proteinase K (20 ug/µl). The eluted DNA samples were incubated at 65°C over-night to reverse crosslink and DNA was recovered with a PCR purification kit (Qiagen) in 50 µl of elution buffer. ChIP-PCR was performed on a QuantiStudio three thermal cycler (Applied Biosystems) from 1.5 ul of ChIP DNA in a total volume of 20 ul containing Fast SYBR Green Master Mix (Applied Biosystems, Cat. 4385612) and 400 nM of each primer

(primer sequences are located in *Supplementary file 7*). Relative ChIP DNA quantities were calculated with the delta-delta CT method and normalized to input DNA. This experiment was repeated two more times with total n = 3.

## MYOD1 chromatin immunoprecipitation assays within adult skeletal muscle

Given that MYOD1 protein oscillates in a circadian fashion, we collected the gastrocnemius muscle tissues from wildtype 12–14 week old male C57BL/6J every 4 hr for 24 hr and pooled six tissues (one per timepoint) prior to chromatin immunoprecipitation with an established MYOD1 antibody (*Cao et al., 2010*). Approximately 800 mg of adult skeletal gastrocnemius muscle was homogenized in cell lysis buffer (10 mM HEPES, pH7.5, 10 mM MgCl$_2$, 60 mM KCl, 300 mM sucrose, 0.1 mM EDTA, pH8.0, 0.1% Triton X-100, 1 mM DTT, protease inhibitors) containing 1% formaldehyde. Cross-linking was stopped after 30 min by adding glycine to a final concentration of 0.125 M. Nuclei and muscle fibers were separated by centrifugation at 25 g for 5 min. The supernatant was collected, and nuclei were spun down by centrifugation at 800 g for 5 min and re-suspended in 350 µl of RIPA buffer (20 mM Tris, pH8.0, 2 mM EDTA, pH8.0, 0.1% sodium deoxycholate, 0.1% SDS, 1% Triton X-100, 140 mM NaCl). The nuclei preparation was sonicated to shear the DNA to 150–600 bp and then centrifuged at 14,000 g for 15 min. The supernatant was collected and pre-cleared with protein A/G slurry at 4°C for 1 hr. Protein A/G beads were removed by centrifugation and supernatant was split into two equal parts, which will be mixed with either MYOD1 Ab or pre-immune serum and incubated overnight at 4°C. The immune-complex was incubated with protein A/G beads slurry for 4 hr and the beads will be washed with RIPA buffer five times and TE buffer twice. DNA was then eluted twice with 250 µl of elution buffer (1% SDS and 0.1 M NaHCO$_3$) for 20 min and supernatant was collected after 5 min centrifugation at high speed. 25 µl of 4 M NaCl was mixed with the 500 µl of elutant and incubated at 65°C overnight. Then 10 µl of 0.5 M EDTA, 20 µl of 1 M Tris pH 6.5 and 2 µl 10 mg/ml proteinase K was added to each sample and incubated at 55°C for 1.5 hr. DNA was recovered with a PCR purification kit (Qiagen). ChIP DNA samples from the MYOD1 pull-down were prepared for Solexa sequencing according to the Illumina protocol using methods described in *Cao et al. (2010)*, with DNA fragments ranging in size from 150 to 600 bp.

## ChIP-Seq analysis

We utilized the FastQ Groomer package to convert raw Fastq reads to Fastq-Sanger format, and then Trimmomatic was used to cut adapter and Illumina-specific sequences from the reads. We next aligned the trimmed reads to the mouse mm10 genome (provided by UCSC database) with the bowtie2 package. Samtools was utilized to convert between sam format to bam format, and then bamToBed was utilized to convert the reads into Bed format. We then utilized the HOMER software package for the remainder of our ChIP-seq processes. Briefly, we utilized the Homer package 'make-TagDirectory' to create tag directories. Due to the the lack of an pre-immune serum control for these studies we used stringent criteria for peak identification. To identify MYOD1 binding sites we utilized the 'findPeaks' package with the following parameters: style –factor –o auto with local background filtering set to 6-fold higher than the average surrounding 10kbp (-L 6). To identify peaks associated with gene promoters we utilized the 'annotatePeaks.pl' package with the mm10 mouse genome. The package 'makeUCSCfile' was used to create a bedgraph formatted file to visualize MYOD1 binding peaks with the UCSC genome browser (mm10 genome). To identify MYOD1 binding sites that are associated with known circadian genes in skeletal muscle, we queried the skeletal muscle circadian transcriptome data downloaded from Circadb (http://circadb.hogeneschlab.org/). We utilized a JTK_CYCLE circadian p-value cutoff of 0.03 for identifying circadian transcripts in skeletal muscle. GEO accession number for raw ChIP-Seq data: GSE122082.

## Gene-Ontology enrichment analysis

The findGO.pl package within HOMER was used to perform functional enrichment analyses of provided gene lists in the form of Entrez gene IDs, with organism set to 'mouse'.

## Plasmids

Clock gene promoters were cloned into pGL3 basic vector based on MYOD1 binding sites identified by MYOD1 ChIP Seq. *Bmal1* promoter (−400 to +154) was amplified from mouse genomic DNA and cloned into pGL3 basic vector and was a gift from Dr. John Hogenesch, Cincinnati Children's Hospital Medical Center (*Sato et al., 2004*). The wild-type expression vectors used in this study include: *MyoD1, Bmal1, Clock*. The *Bmal1*mut vector (Bmal1-R91A) was a gift from Dr. Takahashi from the University of Texas Southwestern (*Hosoda et al., 2004*). The *Clock*mut vector was generated by cloning the mutant Clock$^{Delta19}$ sequence (*King et al., 1997*) into pcDNA3.1. The MYOD1mut (MYOD1Δ3–56) construct was a gift from Dr. Rudnicki at the Ottawa Hospital Research Institute (*Perry et al., 2001*). The control plasmid we used as an empty vector control was the pGEM vector from Promega. As the pRLnull responds highly to MYOD1 expression in C2C12 myotubes, we mutated the responsive E-box and we used the mutant form of pRLnull to normalize transfection. The *Bmal1P*-Luc reporter contains the wiltype *Bmal1* promoter sequence starting 394 bp upstream of transcription start (TSS) and ending 154 bp downstream of the TSS and was a gift from Dr. Takahashi (*Baggs et al., 2009*). The *Tcap* luciferase promoter-reporter construct (*TcapP*-Luc) was a gift from Dr. Davie from Southern Illinois University School of Medicine and contains a 421 base pair promoter fragment that spans from −421 to +1 of the translational start site (*Zhang et al., 2011*). BOS-EF1a-cVenus210-matr3 and BOS-EF1a-nVenus-matr3 were gifts from Dr. Edgardo Rodriguez-Lebron at University of Florida. For Bimolecular fluorescence complementation study, *Bmal1, Bmal2, Clock* and *MyoD1* genes were cloned into BOS-EF1a-cVenus210, downstream of C terminus fragment of venus protein(VC), or cloned into BOS-EF1a-nVenus 210, downstream of N terminus fragment of venus protein (VN). pcDNA3.1 HA VC, pcDNA3.1 HA VN, pcDNA3.1 HA-HDAC5-VN and pcDNA3.1 HA-MEF2C-VC are gift from from Dr. Olivier Kassel from Karlsruhe Institue of Technology of Germany. For Bimolecular fluorescence complementation study, *Bmal1, Bmal2, Clock and MYOD1* genes were either cloned into pcDNA3.1 HA linker VN (venus N) vector, upstream of N terminus fragment of *Venus* protein, or cloned into pcDNA3.1 HA linker VC (venus C) vector, upstream of C terminus fragment of *Venus* protein. All cloned constructs were confirmed by DNA sequencing.

## Western blot

C2C12 myoblasts were obtained from ATCC. Myoblasts were transiently transfected with 150 ng of either pGEM or MYOD1 expression vectors and cells were placed into fresh GM for 24 hr (*Andrews et al., 2010*). Following the 24 hr incubation, myoblasts were washed with ice-cold PBS and then lysed with RIPA buffer. 50 ug of protein from whole-cell lysates were loaded into each well for SDS-PAGE (10%). Proteins were transferred with a semidry apparatus onto PVDF membrane and then blotted with anti-BMAL1 Ab (Sigma SAB4300614, rabbit source, 1:1000) or anti-tubulin (Sigma T6557, mouse source, 1:1000) antibodies. Secondary Ab F(ab')2-goat anti-rabbit IgG (H + L) HRP or F(ab')2-goat anti-mouse IgG (H + L) HRP (1:10000) were applied and the blot was visualized by using Bio-Rad imager. Each lane represents a separate independent experiment (n = 3 biological replicates). BMAL1 bands were quantified with ImageJ software and normalized to the tubulin loading control.

## Site-directed mutagenesis

The QuikChange II Site-Directed Mutagenesis Kit was used to mutate the E-box element in the proximal promoter region of the *TcapP*-Luc, *Bmal1P*-Luc vector and pRLnull according to the manufacturer's instructions. The forward and reverse primer sequences are located in *Supplementary file 8*. The E-box CAGGTG is located in *pRL null* T7 promoter, at −59 to −64 of T7 promoter transcription start site. Genetic mutations were confirmed by DNA sequencing.

## Isolation of skeletal muscle primary cells

Skeletal muscle primary cells were isolated as previously described (*Liu et al., 2015*). Briefly, hindlimb muscles from 8 week old from 2 C57/Bl6J mice were collected and minced in wash media containing Ham's F-10 supplemented with 10% fetal bovine serum (FBS) and 1% Streptomycin/Penicillin. Muscle tissues were then incubated in 800 U/ml Collagenase type II solution (Worthington, Cat. LS004176) at 37°C for 1 hr with gently agitation. After centrifugation, pellets were resuspended in wash media supplemented with 1000 U Collagenase type II and 11 U Dispase (Gibco, Cat. 17105–

041), and incubated at 37°C for 30 min with gently agitation. Next, samples were triturated with a 20-gauge needle, centrifuged and resuspended in wash media. Cell suspensions were filtered using a 40 µm nylon cell strainer (Fisher Scientific, Cat. 22363547), centrifuged, and resuspended in Ham's F-10 supplemented with 20% FBS, 1% Streptomycin/Penicillin, and 2.5 ng/ml basic fibroblast growth factor (bFGF, PeproTech, Cat. 100-18B-100UG). Cells were pre-plated for 30–40 min in an uncoated dish. Unattached cells were transferred to ECM-coated (ECM Gel from Engelbreth-Holm-Swarm murine sarcoma, Sigma, E1270) dishes. Plates were incubated at 37°C, 5% $CO_2$. Media was replaced until day three after plating, thereafter, media was replaced every 2–3 days.

## Dual-Luciferase assay

Luciferase assays were performed from lysates of C2C12 myotubes cell lines (*Andrews et al., 2010*) or skeletal muscle primary myotubes. In each reaction 50 ng of Luciferase reporter plasmid and 150 ng of expression vectors were transfected with X-tremeGENE 9 (Roche) into C2C12 myoblasts or skeletal muscle primary myoblasts in 24-well plates (50,000 cells per well). 5 ng pRL null plasmid was used as a transfection control for C2C12s, while 6 ng pRL null plasmid were used with the skeletal muscle primaries. C2C12 myoblasts were transfected immediately after trypsin digestion, while skeletal muscle primary myoblasts were transfected 24 hr post seeding. After incubation for 24 hr, cells were switched into growth medium (GM) for an additional 24 hr before being switch into differentiation medium (DM) for another 24 hr. Luciferase activity was measured by the Promega Dual Luciferase assay system. *Firefly* luciferase activity was normalized to *Renilla* luciferase activity. The sample size for each experiment reflects n = 3–5 biological replicates per group based on previous data (*Zhang et al., 2012*). Luciferase activity for each transfection are plotted as average fold-change in relation to the pGEM empty vector control ±SEM. Results were analyzed using one-way ANOVA with p set at 0.05.

## Real-time bioluminesence recordings

Co-transfections were performed as described in the previous sections. Approximately 300 ng of luciferase reporter vectors were transfected per 35 mm dish. Expression plasmids or negative controls (pGEM) were co-transfected with approximately 500 ng per 35 mm dish. C2C12 myoblasts and skeletal muscle primary myoblasts were plated with the at 300,000 cells per 35 mm dish. C2C12 myoblasts were transfected at the time of plating with transfection/DNA mix, incubated in GM for 48 hr, and then switched to DM to induce differentiation for 4-full days (DM refreshed every other day). Skeletal muscle myoblasts were incubated with the transfection mix for 24 hr then incubated in fresh GM for an additional 24 hr prior to being switched into DM media for 3-full days. After differentiation, the myotubes (C2C12s and primaries) were synchronized with dexamethasone (1µM) for 90 min, and then washed with 1xPBS twice and 1.5 mL of Recording buffer was added (DMEM-Phenol Red free, 10 mM HEPES, luciferin 0.1 mM, 1% Pen/Strep, 2% Horse Serum, sodium bicarbonate 350 mg/L (*Zhang et al., 2012*). Cells were then air-tight sealed and placed in the Lumicycle apparatus. A 32-well turn table is fully automated so that the photomultiplier tubes captured and recorded the number of photons given off by each culture dish at ten minute intervals. The luminometry data was stored as photon counts/second on a computer using the Lumicycle software by Actimetrix. Circadian statistics were performed with JTK_CYCLE analysis (*Hughes et al., 2010*) R-package to determine circadian amplitude. The sample size for each experiment reflects n = 3–5 biological replicates per group based on previous data (*Zhang et al., 2012*). Average amplitude data were obtained and results were analyzed using one-way ANOVA.

## Bimolecular fluorescence complementation assay (BiFC)

We followed procedures modified from *Kemler et al. (2016)*. To visualize the interaction between transcription factors used a protein harboring *Venus* protein N-terminal fragments and proteins harboring *Venus* protein C-terminal fragments, 250 ng of each of these two plasmids were transiently co-transfected with 100 ng mCherry using X-tremeGENE 9 DNA transfection reagent (Roche) into C2C12 myoblasts. C2C12 density was $2 \times 10^4$ cell per well of eight-well chamber slide (Ibidi, Martinsried, Germany). Fluorescence was visualized using a Leica SP8 confocal microscope 24 hr post transfection.

## In vivo plasmid electroporation and biolumensence recording

Electroporation of tibialis anterior (TA) muscles was conducted as described previously with modification (*Senf and Judge, 2012*). Mice were acutely anesthetized with isoflurane gas (5%, induction; 3% maintenance) delivered via a nose cone. Each TA muscle was injected, through the skin, with 20 µl of 1xPBS containing the combined plasmid constructs. The plasmids were 10 µg of either MYOD1 expression plasmid (right TA) or pGEM control plasmid (left TA), 2 µg of *Renilla* luciferase plasmid, and 20 µg of the *Bmal1P*-Luc reporter. Following injection, electric pulses were delivered using an electric pulse generator (Electro Squareporator ECM 830; BTX, Hawthorne, NY, USA) by placing conducting gel and paddle-like electrodes on the skin on either side of the muscle. Five pulses were delivered in 200 ms interpulse intervals, each with an effective intensity of 50 V/cm and 20 ms duration, as previously used (*Reed et al., 2012*). These electroporation parameters have been shown not induce muscle damage (*Schertzer et al., 2006*; *Schertzer and Lynch, 2008*). Four days following electroporation, TA muscles were collected, homogenized in passive lysis buffer, and luminescence was recorded. Differences in the ratio of *Firefly* luciferase (*Bmal1P*-Luc) luminescence to *Renilla* luciferase luminescence for MYO1/*Bmal1P*-Luc/pRL null (right TA) vs pGEM/*Bmal1P*-Luc/pRL null (left TA) were assessed using a Mann-Whitney non-parametric t-test (Prism8, GraphPad). One outlier was removed from each group based upon the Robust regression and outlier removal (ROUT) test, with a false discovery rate of <0.01.

## Bmal1 siRNA knockdowns

C2C12 myoblasts were plated at a density of 60,000 cells/well in 12-well plates. Cells were transfected with 30 nM *Bmal1* siRNA (Santa Cruz Biotech, Cat. sc-38166) or 30 nM control non-targeted siRNA-A (Santa Cruz Biotech, Cat. sc-37007) using Lipofectamine 3000 as transfection reagent. After incubation for 24 hr, media was replaced with fresh GM for an additional 24 hr before being switched into DM for another 24 hr. Total RNA was extracted using Trizol Reagent (Ambion) and purified using the RNeasy mini kit (Qiagen, Cat. 74106). 1000 ng of total RNA was reverse transcribed using SuperScript VILO cDNA synthesis master mix (Invitrogen, 11754–050). RT-PCR was performed on a QuantiStudio three thermal cycler (Applied Biosystems) from 20 ng of cDNA in a total volume of 20 µl containing Fast SYBR Green Master Mix (Applied Biosystems, Cat. 4385612) and 400 nM of each primer (*Supplementary file 9*). *Rpl26* was used as reference gene for normalization, and relative mRNA expression values were calculated with the delta-delta CT method ($2^{-\Delta\Delta Ct}$).

## Acknowledgements

We thank Dr. Stephen Tapscott (Fred Hutchinson Cancer Research Center, Seattle Washington) for providing the MYOD1 antibody and sequencing support for the ChIP seq experiments. We also thank Dr. JB Hogenesch (Cincinnati Children's Hospital) for providing the skeletal muscle data for our filtering of muscle circadian genes from the CircaDB site. Dr. JS Takahashi (University of Texas Southwestern) Dr. M Rudnicki (Ottawa Hospital Research Institute), Dr. O Kassel (Karlsruhe Institute of Technology of Germany) and Dr. E Rodriguez-Lebron (University of Florida) were all helpful with provision of plasmids for this project. Thanks to Dr. Ximei Cao for technical support with experiments. This work was supported by NIH award R01AR066082 and start up funds from the University of Florida to KAE.

## Additional information

### Competing interests

Yi Cao: Is affiliated with Genentech Inc. The author has no other competing interests to declare. The other authors declare that no competing interests exist.

### Funding

| Funder | Grant reference number | Author |
|---|---|---|
| National Institutes of Health | R01AR066082 | Karyn A Esser |

University of Florida                                    Karyn A Esser

The funders had no role in study design, data collection and interpretation, or the decision to submit the work for publication.

## Author contributions

Brian A Hodge, Conceptualization, Data curation, Formal analysis, Methodology, Writing—original draft, Writing—review and editing; Xiping Zhang, Data curation, Formal analysis, Validation, Investigation, Methodology, Writing—original draft, Writing—review and editing; Miguel A Gutierrez-Monreal, Data curation, Investigation, Methodology, Performed all the primary cell culture transfection experiments required for the revisions; Yi Cao, Resources, Data curation, Methodology; David W Hammers, Data curation, Investigation, Visualization, Methodology, Writing—review and editing; Zizhen Yao, Ping Du, Data curation, Methodology; Christopher A Wolff, Data curation, Formal analysis, Worked with the new in vivo electroporation studies to generate needed data in mice; Denise Kemler, Conceptualization, Methodology, Writing—review and editing; Andrew R Judge, Data curation, Investigation, Methodology, Provided the resources for the in vivo electroporation and performed the injections for the studies requested by the reviewers; Karyn A Esser, Conceptualization, Resources, Data curation, Formal analysis, Funding acquisition, Validation, Investigation, Visualization, Project administration, Writing—review and editing

## Author ORCIDs

Karyn A Esser (iD) http://orcid.org/0000-0002-5791-1441

## Ethics

Animal experimentation: This study was performed in strict accordance with the recommendations in the Guide for the Care and Use of Laboratory Animals of the National Institutes of Health. All of the animals were handled according to approved institutional animal care and use committee (IACUC) protocols (IACUC Study 201809136) of the University of Florida.

## Decision letter and Author response

Decision letter https://doi.org/10.7554/eLife.43017.025
Author response https://doi.org/10.7554/eLife.43017.026

# Additional files

## Supplementary files

• Supplementary file 1. MYOD1 ChIP-Seq genomic binding sites.
DOI: https://doi.org/10.7554/eLife.43017.012

• Supplementary file 2. MYOD1 ChIP-Seq enriched biological processes.
DOI: https://doi.org/10.7554/eLife.43017.013

• Supplementary file 3. List of skeletal muscle circadian genes bound by MYOD1.
DOI: https://doi.org/10.7554/eLife.43017.014

• Supplementary file 4. Expression of 536 circadian MYOD1 target genes in MYOD1-CE muscle tissue.
DOI: https://doi.org/10.7554/eLife.43017.015

• Supplementary file 5. Bioinformatics approach to identify *Tcap* as a muscle-specific MYOD1 and BMAL1 target gene (associated gene-lists).
DOI: https://doi.org/10.7554/eLife.43017.016

• Supplementary file 6. Primer sequences used in RT-PCR of adult skeletal muscle.
DOI: https://doi.org/10.7554/eLife.43017.017

• Supplementary file 7. Primer sequences used in site-directed mutagenesis.
DOI: https://doi.org/10.7554/eLife.43017.018

• Supplementary file 8. Primer sequences used in ChIP-PCR.
DOI: https://doi.org/10.7554/eLife.43017.019

• Supplementary file 9. Primer sequences used in RT-PCR in C2C12 myotubes.
DOI: https://doi.org/10.7554/eLife.43017.020

• Transparent reporting form
DOI: https://doi.org/10.7554/eLife.43017.021

### Data availability

ChIP seq data for muscle with MYOD is deposited in GEO under accession code GSE122082.

The following dataset was generated:

| Author(s) | Year | Dataset title | Dataset URL | Database and Identifier |
|---|---|---|---|---|
| Hodge BA, Zhang X | 2018 | MYOD ChIPseq and skeletal muscle tissue | https://www.ncbi.nlm.nih.gov/geo/query/acc.cgi?acc=GSE122082 | NCBI Gene Expression Omnibus, GSE122082 |

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
