## [Decision Letter]

Thank you for submitting your article "MYOD1 functions as a clock amplifier as well as a critical co-factor for downstream circadian gene expression in muscle" for consideration by *eLife*. Your article has been reviewed by three peer reviewers, one of whom is a member of our Board of Reviewing Editors, and the evaluation has been overseen by Kevin Struhl as the Senior Editor. The following individual involved in review of your submission has agreed to reveal his identity: James Ryall (Reviewer #3).

The reviewers have discussed the reviews with one another and the Reviewing Editor has drafted this decision to help you prepare a revised submission.

Summary:

The manuscript from Hodge et al. addresses the exciting field of circadian rhythms. Specifically asking how the clock is regulated in a tissue specific manner. To date, numerous manuscripts have shown an important role for circadian rhythms, however few have addressed the regulators of the rhythm itself. In this manuscript the authors demonstrate that *MyoD1* regulates the amplitude of the clock and that *MyoD1* regulates a muscle specific gene, *Tcap* in a clock dependent manner. This work is exciting and timely. In addition, the experiments are performed to a high standard with appropriate conclusions. There two major concerns that the authors need to address; 1) the overreliance of C2C12 cells, 2) the binding/recruitment between BMAL1, MYOD1 and *TCap*.

Essential revisions:

1) The major limitations of this study are related to the sole use of C2C12 cell line and overexpression of MYOD1 and clock regulators. In addition the effects of MYOD1 and BMAL1:CLOCK are evaluated on transfected luciferase reporters. Performing any of the in vitro experiments in primary myogenic cells would suffice.

2) Primary myoblasts derived from MYOD1 knock-out experiments could be employed to validate some of the findings reported here. Do the 536 clock genes identified as MYOD1 lose circadian regulation in MYOD1^-/-^ primary myoblasts? Does reconstitution with MYOD1 restore their circadian expression? Expression of a small subset of the 536 genes could be evaluated by RT-qPCR.

3) My major query for the authors is why the *MyoD1* ChIPseq did not identify *Tcap* as a direct target and yet in vitro luciferase experiments demonstrated a clear response to *MyoD1* alone? This is worrisome as it could suggest that the in vitro studies utilizing an immortalized cell line do not represent the in vivo condition, especially as *Tcap* has only previously been found to be regulated by *MyoD1* in vitro (Zhang et al., 2011). Another possible explanation for this discrepancy could be due to the use of proliferating cells (versus terminally differentiated muscle for the ChIPseq). It would be useful for the investigators to determine whether their model holds true in differentiated C2C12 cells and/or primary cells. Alternatively, evidence of circadian *Bmal1* binding to the Ebox regions of *Tcap* in skeletal muscle (via ChIP-PCR) would provide strong support for the hypothesis.

4) Is BMAL1 co-recruited at a subset of the 536 MYOD1-target circadian genes by *Bmal1* ChIP assay? Does MYOD1 interact with the non-canonical E-box at the *Bmal1* and tandem E-box *Tcap* regions? These data should already be available to the authors.

5) Electroporation of *Bmal1-Luc* and *Tcap-Luc* (WT and mutant constructs) in skeletal muscle of control and MYOD1 knock-out mice would allow to convincingly address the role of MYOD1 in regulating circadian regulation of BMAL1 and *Tcap* via the identified E-boxes in a physiological setting.

---

## [Author Response]

Essential revisions:1) The major limitations of this study are related to the sole use of C2C12 cell line and overexpression of MYOD1 and clock regulators. In addition the effects of MYOD1 and BMAL1:CLOCK are evaluated on transfected luciferase reporters. Performing any of the in vitro experiments in primary myogenic cells would suffice.

To address the concerns with our reliance on C2C12 myotubes, we added new in vitro experiments in which we use primary muscle cells isolated from mouse hindlimb muscles. This new data has been added to Figures 2, 3 and 6 and compliments our initial findings in C2C12 myotubes. We performed a dual-luciferase reporter assay in a similar fashion to the experiments performed in C2C12 myotubes (Figure 2C) and found that cotransfection with the wildtype MYOD1 expression vector, but not the MYOD1 mutant vector, was sufficient to significantly transactivate the *Bmal1*-Luc reporter in primary myotubes (Figure 2D). Also in Figure 2 we include new data in which we used in vivo electroporation to demonstrate that the *Bmal1* promoter is strongly transactivated by MYOD1 in tibialis anterior muscles of mice. We also performed real-time bioluminescence recordings in synchronized myotubes generated from primary muscles cells. In Figure 3F we demonstrate that the wildtype *Bmal1P*-Luc reporter oscillated in a circadian fashion, while the *Bmal1Pmut*-Luc reporter failed to oscillate with the same amplitude. In Figure 6G we show that synchronized primary myotubes co-transfected with wildtype *TcapP*-Luc display a greater bioluminescence amplitude than those transfected with the mutant *TcapP-E2-5’*-Luc reporter. Taken together these data with the use of primary cells and in vivo electroporation all consistently reinforce our findings originally obtained in C2C12 myotubes.

The second concern in this point was about reliance on overexpression strategies and we agree that our initial submission relied heavily on the use of over-expression plasmids for gain/loss of function analysis and promoter-reporter luciferase reporters to assay gene regulation. In our revised submission we have provided additional in vivo data to complement our promoter:reporter findings. Firstly, we have evaluated circadian gene expression via RT-PCR analysis in adult muscle tissue from MYOD1-CE mice and wildtype controls. MYOD1CE mice were chosen for our study given that they have constitutively low levels of *MyoD1* expression and could therefore serve as an in vivomodel of *MyoD1* knockdown. The MYOD1-CE mice been used by my lab in our paper Zhang et al., 2012, and the original reference for the mice is: Chen and Goldhamer, 2004.

From these analyses we found that circadian genes that are targeted by MYOD1 in our ChIP-seq analysis (*Ppp1R3c, Asb2, Nrip1*, and *Vegfa*) displayed significant reductions in expression with marked alterations to their circadian gene expression profile (Figure 1). Additionally, we found that circadian expression of *Tcap* mRNAis lost in MYOD1-CE muscle and this adds to the data in Figure 5. We also performed siRNA mediated knockdown of *Bmal1* in C2C12 myotubes and found a significant decrease in expression of endogenous *Tcap* and *MyoD1* mRNA levels (these data are now Figure 5—figure supplement 1).

2) Primary myoblasts derived from MYOD1 knock-out experiments could be employed to validate some of the findings reported here. Do the 536 clock genes identified as MYOD1 lose circadian regulation in MYOD1^-/-^ primary myoblasts? Does reconstitution with MYOD1 restore their circadian expression? Expression of a small subset of the 536 genes could be evaluated by RT-qPCR.

Our MYOD1 ChIP-seq allowed us to identify 536 circadian genes that are direct targets of MYOD1 in skeletal muscle. We agree with the reviewers that MYOD1 loss of function experiments would help to validate our claim that MYOD1 regulates the circadian expression of these genes. To address this issue we decided to utilize the MYOD1-CE mice for two reasons. Scientifically these mice have been previously characterized as having low *Myod1* expression in adult skeletal muscle compared to wildtype control mice. Pragmatically, we have the MYOD1-CE mice in our breeding facility at UF but we do not maintain the MyoD^-/-^ mice. To obtain the mice, get through quarantine and breed them up would have pushed these experiments well beyond the 2 month deadline provided by *eLife*.

For this manuscript we collected gastrocnemius muscles over a circadian timecourse from wildtype and MYOD1-CE mice. We chose a subset of potential candidate genes from the list of the 536 circadian genes targeted by MYOD1 in our ChIPseq experiment. The genes we chose to analyze included robust circadian genes and are the following: *Ppp1r3c, Vegfa, Asb2*, and *Nrip1*. We performed RT-qPCR on the RNA to determine if their circadian pattern and/or magnitude of expression were altered in the muscle of the MYOD1 knockdown mouse (MYOD1-CD). We found that all 4 genes displayed significant alterations in circadian mRNA expression patterns as indicated by JTK_CYCLE analysis (data now in Figures 1C-G).

3) My major query for the authors is why the MyoD1 ChIPseq did not identify Tcap as a direct target and yet in vitro luciferase experiments demonstrated a clear response to MyoD1 alone? This is worrisome as it could suggest that the in vitro studies utilizing an immortalized cell line do not represent the in vivo condition, especially as Tcap has only previously been found to be regulated by MyoD1 in vitro (Zhang et al., 2011). Another possible explanation for this discrepancy could be due to the use of proliferating cells (versus terminally differentiated muscle for the ChIPseq). It would be useful for the investigators to determine whether their model holds true in differentiated C2C12 cells and/or primary cells. Alternatively, evidence of circadian Bmal1 binding to the Ebox regions of Tcap in skeletal muscle (via ChIP-PCR) would provide strong support for the hypothesis.

We appreciate this reviewer’s concern about use of proliferating vs. differentiated cells but one thing we need to clarify is that all the transfection data in this manuscript come from experiments performed on differentiated myotubes or muscle tissue and not myoblasts. We have gone through the manuscript to make sure that point is very clear as we agree that what is detected in myoblasts does not always correspond to the differentiated myotube/muscle tissue. In addition, as noted above we now include a number of new experiments in which we used primary muscle cells differentiated into myotubes as well as in vivo muscle tissue to strengthen the study.

The other concern noted is that we did not detect MYOD1 binding on the Tcap promoter in our ChIPseq data set. This was perplexing to us but between our transfection outcomes coupled with our BiFC data and the results from a different lab that showed MYOD1 binding within the *Tcap* promoter via ChIP-PCR during embryonic muscle development we felt that maybe MYOD1 might be more functioning as part of a complex that could get disrupted in the ChIP preparation. For the revisions, we went back to muscle tissue and worked on the crosslinking conditions and performed targeted ChIP-PCR to directly test if MYOD1 does indeed target the *Tcap* promoter. We included primers for both the Tandem E-box element as well as the E1 E-box element. With this approach we found a significant enrichment for MYOD1 binding within the tandem E-box (Figure 6C) but no significant enrichment of MYOD1 for the E1 E-box (Figure 6A). Additionally, we also have included data showing strong BMAL1 binding enrichment for the tandem E-box (Figure 6D) and the E1 E-box (Figure 6B).

4) Is BMAL1 co-recruited at a subset of the 536 MYOD1-target circadian genes by Bmal1 ChIP assay? Does MYOD1 interact with the non-canonical E-box at the Bmal1 and tandem E-box Tcap regions? These data should already be available to the authors.

We agree that demonstrating that BMAL1 also targets the subset of 536 MYOD1-target circadian genes would greatly strengthen the interpretation that BMAL1 is co-recruited to these genes by MYOD1 but it is beyond the scope of our time to do *Bmal1* ChIPseq within a 2 month window. However, we did query a recently published BMAL1 ChIP-seq dataset also performed in adult skeletal muscle (Dyar et al., 2018). Of the two BMAL1 ChIP-seq biological replicates we found that there was overlap of approximately 48% to 78% of the 536 MYOD1-target circadian genes. While the range was quite variable between the two Bmal1 ChIPseq replicates there seems to be around 50% or more coverage with both MYOD1 and BMAL1 supporting our concept that this mechanism goes beyond regulation of Tcap.

5) Electroporation of Bmal1-Luc and Tcap-Luc (WT and mutant constructs) in skeletal muscle of control and MYOD1 knock-out mice would allow to convincingly address the role of MYOD1 in regulating circadian regulation of BMAL1 and Tcap via the identified E-boxes in a physiological setting.

We agree with this reviewer and we provide new in vivodata showing that the *Bmal1P*-Luc reporter is transactivated by MYOD1 in adult skeletal muscle. For these experiments we electroporated the tibialis anterior muscle with *Bmal1P*-Luc with MYOD1 or +pGEM(control). The magnitude of *Bmal1* activation is similar to what we found with our in vitro studies (new data are included in Figure 2E).

To address *Tcap* regulation in vivo, we quantified *Tcap* mRNA levels over a circadian time-course in MYOD1CE skeletal muscle, and found a complete loss of circadian expression (Figure 6G, H). These data indicate that MYOD1 expression influences the expression and amplitude of both the core molecular clock as well as downstream circadian genes in skeletal muscle.